# Beyond Rational Illusion: Behaviorally Realistic Strategic Classification

Xinpeng Lv [1]   Yunxin Mao [1]   Renzhe Xu [2]   Chunyuan Zheng [3]   Yikai Chen [1]   Haoxuan Li [3]   Yang Shi [4]
Jinxuan Yang [5]   Yuanlong Chen [6]   Yuanxing Zhang [4]   Shaowu Yang [1]   Wenjing Yang [1]   Zhouchen Lin [3]
Haotian Wang [1]

## Abstract

Strategic classification (SC) studies the interaction between decision models and agents who strategically manipulate their features for favorable outcomes. Existing SC frameworks typically rely on the idealized assumption that agents are strictly rational. However, evidence from behavioral economics and psychology consistently shows that real-world decision-making is often shaped by cognitive biases, deviating from pure rationality. To formalize this limitation, we identify and define a new problem setting, termed the *behaviorally realistic strategic classification problem*, where agents' strategic manipulations deviate from full rationality due to psychological biases. Motivated by the identified limitation, we propose the **Prospect-Guided Strategic Framework** (Pro-SF) to address the problem, a principled framework grounded in prospect theory to model and learn under behaviorally realistic strategic responses. Specifically, to capture behaviorally realistic strategic manipulations, our framework reformulates the Stackelberg-style interaction between agents and the decision-maker by incorporating three key mechanisms inspired by prospect theory, including the asymmetry between benefits and costs, different subjective reference points, and non-rational probability distortion. Experiments on synthetic and real-world datasets establish Pro-SF as a behaviorally grounded ap-

proach to strategic classification, bridging machine learning and behavioral economics for more reliable deployment in the real world.

## 1. Introduction

Machine learning models are playing an increasingly critical role in diverse human-serving domains, such as hiring (Sánchez-Monedero et al., 2020), credit scoring (Jagtiani & Lemieux, 2019), and college admissions (Kuvcak et al., 2018). In such settings, individuals may strategically modify their observable features to obtain favorable decisions. This phenomenon reflects Goodhart's Law (Strathern, 1997): "*When a measure becomes a target, it ceases to be a good measure*." Once a model's decision rule becomes known or anticipated, agents often engage in 'gaming' behaviors to manipulate outcomes. For example, a loan applicant might temporarily inflate their reported income to appear more creditworthy. To anticipate such manipulations, strategic classification (SC) (Hardt et al., 2016) provides a specialized machine learning (ML) framework by considering the Stackelberg-style interaction between decision makers and strategic agents (Ghalme et al., 2021; Singh & Kulkarni, 2024; Chen et al., 2020), serving as a key bridge between ML and social sciences.

Given its growing influence, it is increasingly essential to critically examine the foundational principles of SC. To be specific, the framework of SC rests on a simplified assumption that *agents are perfectly rational* (Hardt et al., 2016; Milli et al., 2019). Consequently, under this view, an agent modifies features if, and only if, the expected benefit outweighs the cost. However, like a double-edged sword, the assumption of fully rational agents might result in conflicts with realistic scenarios, as individuals often behave in ways that deviate sharply from rational utilities:

- **Example 1.** In financial investment (Banerji et al., 2020), individuals often react more strongly to a potential \$80 loss than to a potential \$100 gain of equal probability. The same magnitude of outcome thus triggers disproportionate behavioral responses.

- **Example 2.** In credit scoring (Banerji et al., 2020), con-

---

[1]College of Computer, National University of Defense Technology, Changsha, China [2]Institute for Theoretical Computer Science, Shanghai University of Finance and Economics, Shanghai, China [3]State Key Lab of General AI, School of Intelligence Science and Technology, Peking University, Beijing, China [4]School of Computer Science, Peking University, Beijing, China [5]Faculty of Engineering, University of Sydney, Sydney, Australia [6]Faculty of Computing, Harbin Institute of Technology, Harbin, China. Correspondence to: Haotian Wang <wanghaotian13@nudt.edu.cn>.

*Proceedings of the 43rd International Conference on Machine Learning*, Seoul, South Korea. PMLR 306, 2026. Copyright 2026 by the author(s).

*Figure 1.* Illustrative real-life scenarios of behavioral biases: (a) financial investment shaped by loss aversion, (b) credit scoring influenced by reference bias, and (c) disease screening affected by probability distortion.

sider loan approval requires applicants to exceed a threshold $A$. Those whose subjective reference point $B$ is just below $A$ tend to make marginal adjustments, whereas those far below the threshold usually forgo effort.

- **Example 3.** In disease screening (Dwyer et al., 2022), consider a rare disease with a true prevalence of only 0.5%. Although the objective probability of infection is negligible, some individuals persist in seeking repeated testing, subjectively inflating the small chance of illness.

Unfortunately, the non-rational behaviors characterized in the three examples, i.e., asymmetry between the benefit and cost, behaviors based on different reference points, and distorted probability towards events, are not considered in conventional SC frameworks. More broadly, extensive evidence from behavioral economics and psychology shows that real-world decision making systematically deviates from rational optimization (Carroll & Johnson, 1990; Tversky & Kahneman, 1992; Jones, 1999; Ariely & Jones, 2008). Therefore, we highlight that the rational-agent assumption, while mathematically convenient, often collapses in real-world settings. This mismatch motivates the need for a refined SC framework to capture such realistic behavioral deviations more precisely, which raises a central research challenge:

> *Can existing SC methods account for strategic manipulations shaped by psychological biases, beyond the conventional rational-utility paradigm?*

In response to this question, we first define **behaviorally realistic strategic classification** *(BR-SC)* as a new problem where agents' manipulations reflect realistic behavioral patterns driven by psychological biases, rather than fully rational best-response rules. Accordingly, the classifier is designed to account for these manipulations, ensuring robustness in deployment. Fortunately, prospect theory (Kahneman & Tversky, 2013) and followed behavioral economics highlight three pervasive mechanisms for realistic behaviors: (1) *Loss aversion*. Individuals tend to weigh potential losses more heavily than equivalent gains (see Fig. 1(a)). (2) *Reference bias*. Individual decision depends on subjective reference points, which differ across individuals based on their personal circumstances, expectations, or prior experiences (Fig. 1(b)). (3) *Probability distortion*. Individuals

overweight small-probability events, showing a tendency to "gamble" on unlikely opportunities (see Fig. 1(c)).

These behavioral mechanisms result in two consequential failure modes for existing SC methods, *over-defense* and *under-defense*, which degrade robustness of decision models for SC (see detailed derivation in Sec. 4). Therefore, we propose the **Prospect-Guided Strategic Framework** *(Pro-SF)* to address the BR-SC problem. Pro-SF introduces a paradigm shift for SC by integrating three key principles (i.e., asymmetry between gains and losses, subjective reference points, and probability distortion) into the Stackelberg game framework of SC. This paradigm shift captures how agents actually manipulate their features in practice and redefines both the dynamics of agents' strategic manipulation and the decision maker's optimization objective.

**Our main contributions are summarized as follows:**

- We identify and formalize the limitations of the rational-agent assumption in strategic classification by introducing a new problem setting, termed *behaviorally realistic strategic classification* (BR-SC). We further characterize two failure modes induced by rational-agent modeling, namely *over-defense* and *under-defense*.

- We propose the **prospect-guided strategic framework** *(Pro-SF)*, which integrates loss aversion, reference bias, and probability distortion into the Stackelberg game framework, providing a more realistic solution for strategic classification.

- We conduct extensive experiments on synthetic and real-world datasets to demonstrate that Pro-SF achieves robust performance across diverse behavioral regimes. Ablation studies and sensitivity analyses further illustrate how each behavioral component contributes to robustness.

## 2. Related Work

### 2.1. Strategic Classification

Strategic classification (SC) studies how individuals manipulate features to obtain favorable decisions (Hardt et al., 2016). Prior work largely focuses on designing decision rules that are robust to such manipulations (Dong et al., 2018; Shavit et al., 2020; Chen et al., 2020; Harris et al., 2021; Zrnic et al., 2021; Tsirtsis et al., 2024; Liu et al., 2024;

Lv et al., 2026a), as well as leveraging strategic behavior to incentivize genuine improvement via causal or action-based formulations (Miller et al., 2020; Chen et al., 2023; Horowitz & Rosenfeld, 2023; Vo et al., 2024; Efthymiou et al., 2026; Chang et al., 2024; Wang et al., 2022; 2025; Yang et al., 2025). Related settings include performative prediction, where repeated deployment alters the data distribution (Perdomo et al., 2020; Rosenfeld et al., 2020; Hardt et al., 2022; Mendler-Dünner et al., 2022; Wang et al.; Lv et al., 2026b), and societal-level regulation of strategic behavior, e.g., optimizing long-term welfare or mitigating demographic disparities (Haghtalab et al., 2020; Estornell et al., 2023a; Xie & Zhang, 2024; Zhang et al., 2022; Estornell et al., 2023b; Keswani & Celis, 2023; Lv et al., 2026c). Recent evidence suggests that human strategic manipulation deviates substantially from the rational assumption (Ebrahimi et al., 2025; Xie et al., 2025), motivating our behaviorally realistic formulation (BR-SC) and the prospect-theoretic framework (Pro-SF). **More related work** on strategic classification is included in Appendix A.1.

### 2.2. Behavioral Economics in Machine Learning

Behavioral economics (Mullainathan & Thaler, 2000) challenges the classical assumption that agents act as perfectly rational utility maximizers, emphasizing instead that choices are systematically shaped by cognitive biases and contextual factors (Tversky & Kahneman, 1992; Ariely & Jones, 2008). A central framework crystallizing these insights is *prospect theory* (Tversky & Kahneman, 1992; Kahneman & Tversky, 2013), which models how individuals evaluate outcomes relative to reference points, exhibit asymmetric sensitivity to gains and losses, and distort probabilities. These principles have influenced multiple domains: in economics and social science, they inform finance, consumer behavior, and public policy (Borkar & Chandak, 2021; Mercer, 2005; Vis, 2011); in computational settings, they shape reinforcement learning (Shen et al., 2014; Prashanth et al., 2016), mechanism design (Kuvcak et al., 2018; Leoneti & Gomes, 2021), and resource allocation (Holmes Jr et al., 2011). **More related work** is included in Appendix A.3.

## 3. Preliminary

We briefly review the strategic classification (SC) paradigm. Random variables are denoted by uppercase letters (e.g., $X$, $Y$), their realizations by lowercase (e.g., $x$, $y$), and boldface for vectors or matrices (e.g., $\mathbf{x}$, $\mathbf{X}$).

### 3.1. Rational Strategic Classification Model

The strategic classification problem is modeled as a Stackelberg game (Li & Sethi, 2017), where a **decision maker** defines a classification function $f : \mathbb{R}^d \to \{-1, 1\}$, and **decision subjects** (agents) strategically manipulate their fea-

tures from $\mathbf{x}$ to $\mathbf{x}'$ at a cost $c : \mathcal{X} \times \mathcal{X} \to \mathbb{R}_{\geq 0}$ (Hardt et al., 2016; Miller et al., 2020). The optimal manipulated feature $\mathbf{x}'$ is determined by the best-response function $b_R(\mathbf{x})$:

**Definition 3.1** (Rational Strategic Manipulation). *The optimal modified feature vector $\mathbf{x}'$ is determined by:*

$$\mathbf{x}' = b_R(\mathbf{x}) = \arg\max_{\mathbf{x}' \in \mathcal{X}} \left[ f(x') - \lambda c(x, x') \right], \quad (1)$$

*where $f(x') \in \{-1, 1\}$ is the classification result after modification, $c(x, x')$ is the manipulation cost, $\lambda > 0$ is a trade-off parameter, and $\mathcal{X}$ is the feature space. Usually, the cost of manipulation is modeled as the Mahalanobis distance (Gavish et al., 2022; Chen et al., 2023).*

From the decision maker's perspective, the classification rule $f$ is designed to remain robust under such strategic manipulation:

**Definition 3.2** (Decision Optimization). *To mitigate manipulation, the decision maker optimizes $f$ to maximize expected accuracy against strategic manipulation:*

$$f^* \in \arg\max_{f \in \mathcal{F}} \mathbb{E}_{(\mathbf{x},y) \sim \mathcal{D}} \left[ \mathbb{1}\left( f(b_R(\mathbf{x})) = y \right) \right], \quad (2)$$

*where $\mathcal{F}$ is the set of all feasible classification rules, $\mathbb{1}$ denotes the indicator function, and $y$ is the observed label.*

### 3.2. The Achilles' Heel of Strategic Classification

The rational-agent assumption underlying Eq. (1) and Eq. (2) is mathematically convenient but behaviorally restrictive. Concretely, a large body of work in behavioral economics and psychology shows that human decision making systematically departs from perfect rationality (Tversky & Kahneman, 1992; Ariely & Jones, 2008; Kahneman & Tversky, 2013; Borkar & Chandak, 2021). Among these, **Prospect Theory** (Tversky & Kahneman, 1992; Kahneman & Tversky, 2013) provides a unifying framework, capturing how individuals evaluate uncertain outcomes. Accordingly, we focus on three particularly relevant deviations from rationality that serve as the foundation for our behaviorally realistic formulation:

- **Loss aversion in manipulation.** Agents subjectively inflate perceived effort (e.g., manipulation cost) relative to gains. Consequently, agents may give up manipulations that would be beneficial under a rational-utility model because the perceived loss outweighs the perceived benefit.

- **Reference bias.** Agents evaluate outcomes relative to a subjective reference point (e.g., an expected outcome) rather than an absolute term. This means the utility of a successful manipulation varies across individuals, violating the rational-utility assumption commonly used in the classical SC paradigm.

- **Probability distortion.** Agents may distort perceived acceptance probabilities, e.g., overweighting small probabilities. As a result, they can misjudge how much manipulation is needed to cross the decision threshold, leading to sub-optimal or insufficient adjustments.

# 4. Theory: Behavioral Mismatch and Failure Modes

We now provide a theoretical analysis of why classifiers trained under the rational-agent assumption can systematically fail in practice due to behavioral mismatch. We then formalize two characteristic failure modes: *over-defense* and *under-defense*.

## 4.1. Behavioral Mismatch and Deployment Bias

Without loss of generality, the agent's strategic manipulation can be summarized by a behavioral response function $b_\theta(\cdot)$ induced by a utility $U_\theta(x, x')$:

$$b_\theta(x) \in \arg\max_{x' \in \mathcal{X}} U_\theta(x, x'), \qquad (3)$$

where $\theta$ denotes a general behavioral parameterization. Let $(X, Y) \sim \mathcal{D}$ and $X' = b_\theta(X)$; the post-manipulation distribution is then defined as $\mathcal{D}_\theta := \text{Law}(X', Y)$, which differs from the original feature distribution.

Classical strategic classification assumes that agents respond by maximizing a rational utility, yielding a rational response $b_R(x)$ and a corresponding distribution $\mathcal{D}_R := \text{Law}(b_R(X), Y)$. This creates a **behavioral mismatch**: the classifier is trained on the post-manipulation distribution $\mathcal{D}_R$ induced by $b_R$, but is deployed under the distribution $\mathcal{D}_\theta$ induced by $b_\theta$. Therefore, we have the following proposition, with a detailed proof in Appendix B.

**Proposition 4.1** (Irreducible deployment bias under behavioral mismatch)**.** *Let $f_R$ denote a classifier optimized on $\mathcal{D}_R$, and define the deployment error $\delta(\theta) := \mathbb{E}_{(x,y) \sim \mathcal{D}_\theta}[\mathcal{L}(f_R(x), y)]$. Suppose $\mathcal{D}_\theta \neq \mathcal{D}_R$, and that this mismatch affects a non-zero mass of samples near the decision boundary, i.e., there exist $\epsilon > 0$ and $\gamma > 0$ such that*

$$\mathbb{P}_{(x,y) \sim \mathcal{D}_\theta}\Big(|s(x) - \tau| \leq \gamma,\ y \neq \mathbf{1}\{s(x) \geq \tau\}\Big) \geq \epsilon. \quad (4)$$

*Then $\delta(\theta) > 0$, i.e., a non-vanishing deployment error persists under the rational-agent assumption.*

Next, we show that behavioral mismatch manifests through two *directional* and practically failure modes: *over-defense* and *under-defense*.

## 4.2. Over-defense

Over-defense arises when a classifier trained under the rational-agent assumption anticipates manipulations that

do not occur in practice (Fig. 2a). Agents who perceive themselves as far from acceptance due to loss aversion or reference bias, may choose not to manipulate, even when manipulation would be optimal under a rational utility.

**Definition 4.2** (Over-defense)**.** *Let $f_R$ denote a classifier trained under the rational-agent assumption. $f_R$ exhibits* over-defense *if it adjusts its decision rule to defend against strategic manipulations that are unlikely to occur at deployment, leading to degraded classification performance.*

**Example 1** (Over-defense in a linear classifier)**.** Consider a linear classifier $f(x) = \text{sign}(w^\top x - \tau)$. Under the rational-agent assumption, it is trained to defend against agents who minimally manipulate their features to cross the decision boundary. If, in practice, some agents do not manipulate at all, this anticipation causes the learned threshold to shift from $\tau$ to $\tau' > \tau$. Consequently, some instances that satisfy $w^\top x \geq \tau$—and would be accepted without any manipulation—are now rejected:

$$w^\top x \geq \tau \quad \text{but} \quad w^\top x < \tau' \ \Rightarrow \ f'_R(x) = -1. \quad (5)$$

This illustrates how defending against nonexistent manipulations induces false negatives.

Over-defense shifts the decision boundary to counter manipulations that do not occur in practice, thereby converting genuinely positive instances into false negatives. We formalize the resulting accuracy degradation in the following proposition (see proof in Appendix C).

**Proposition 4.3** (Accuracy Degradation from Over-defense)**.** *Let $f_R$ be a classifier trained under the rational-agent assumption. Suppose there exists a non-negligible set of agents who abstain from manipulation due to behavioral biases, even though rational agents at the same features would manipulate. If the over-defense-induced boundary shift is sufficiently large relative to the residual training error on this abstention set, then*

$$\mathbb{E}[\mathcal{L}(f_R(x), y)] \ > \ \mathbb{E}[\mathcal{L}(f_R(b_R(x)), y)], \qquad (6)$$

*where $\mathcal{L}$ is the loss function for classification.*

## 4.3. Under-defense

Under-defense occurs when agents manipulate more aggressively than anticipated by a classifier trained under the rational-agent assumption (Fig. 2b). Behaviorally, probability distortion may cause agents to overestimate their chances of acceptance, while loss aversion amplifies the fear of rejection, leading them to overshoot the manipulation level predicted by rational utility maximization.

**Definition 4.4** (Under-defense)**.** *A classifier $f_R$ with the rational-agent assumption exhibits* under-defense *if it defends only against rational manipulations, while agents in practice make larger adjustments that exceed this defense, resulting in degraded classification performance.*

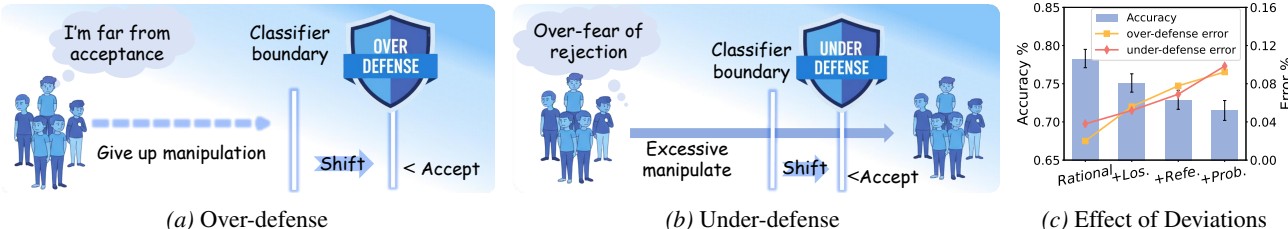

*Figure 2.* Illustration of two failure modes induced by the rational-agent assumption. (a) Over-defense caused by agents giving up manipulation. (b) Under-defense caused by excessive manipulation. (c) Effects of cumulative behavioral deviations on a rational-based classifier (*Los.* = loss aversion, *Refe.* = reference bias, *Prob.* = probability distortion).

**Example 2** (Under-defense in a linear classifier). Consider a linear classifier $f(x) = \text{sign}(w^\top x - \tau)$. Under rational assumptions, agents are expected to manipulate minimally to the decision boundary. If, in practice, some agents over-shoot this endpoint, a classifier trained only against rational manipulations fails to correctly classify these instances:

$$f_R(b_R(x)) = y \quad \text{but} \quad f_R(x^*) \neq y. \tag{7}$$

Under-defense leaves parts of the manipulated feature space unprotected, allowing strategic agents to bypass the intended defense. We formalize the resulting performance degradation in the following proposition (see proof in Appendix D).

**Proposition 4.5** (Accuracy Degradation from Under-defense). *Let $f_R$ be a classifier trained under the rational-agent assumption. Suppose there exists a non-negligible set of agents who overshoot the rational manipulation endpoint due to behavioral biases, landing in regions where $f_R$ provides no defense. If the overshoot-induced disagreement is sufficiently large relative to the residual training error on this overshoot set, then*

$$\mathbb{E}\big[\mathcal{L}(f_R(b_\theta(x)), y)\big] > \mathbb{E}\big[\mathcal{L}(f_R(b_R(x)), y)\big], \tag{8}$$

*where $b_R(x)$ denotes the rational manipulation endpoint and $b_\theta(x)$ the actual manipulated feature under behavioral biases.*

As shown in Fig. 2c, as agents are increasingly influenced by psychological biases, the performance of a classifier trained under the rational-agent assumption deteriorates.

## 5. Prospect-Guided Strategic Framework

### 5.1. Problem Formulation

To better reflect real-world strategic behavior, we formalize a practical strategic classification setting, termed **behaviorally realistic strategic classification problem** *(BR-SC)*. Unlike classical strategic classification, which assumes agents follow rational best-response manipulations, BR-SC allows the manipulations of agents to be influenced by psychological biases.

**Problem 5.1** (Behaviorally Realistic Strategic Classification (BR-SC)). *The BR-SC problem consists of two coupled components:*

- ***Behavioral manipulation model.** Given a data distribution $D$ over $(x, y)$, a cost function $c$, and a classifier $f$, agents are modeled by a behaviorally realistic manipulation function $b_\theta(x)$.*

- ***Classifier learning under behavioral responses.** Based on the behavioral manipulation model $b_\theta(x)$, the decision maker aims to train a classifier $f^* \in \mathcal{F}$ that performs robustly under behaviorally biased strategic manipulations.*

**Example 3** (loan approval). Consider a loan approval system, in the BR-SC problem, applicants may still act strategically, but their decisions are often influenced by psychological biases, such as loss aversion, subjective reference points, and distorted beliefs about approval probability. As a result, strategic responses in real-world deployment systematically deviate from idealized rational best responses, giving rise to a more realistic strategic classification problem.

To address the *BR-SC* problem, we introduce the **Prospect-Guided Strategic Framework** *(Pro-SF)*, which leverages prospect theory to capture realistic agent manipulations and reframe the learning objective for the classifier.

### 5.2. Modeling Prospect-Guided Utility for Agents

We specify an explicit utility-based model for agents' strategic behavior. In particular, we adopt a prospect-guided utility that incorporates three irrational mechanisms: loss aversion, probability distortion, and reference bias.

**Loss aversion in manipulation.** To capture loss aversion in agents' manipulations, we adopt a prospect-style value function that evaluates perceived gains and losses on asymmetric scales. Specifically, given a candidate manipulation $x$, the agent's subjective value is defined as

$$v(x) = v_{\text{gain}}(x)^\alpha - \kappa\,(v_{\text{loss}}(x))^\beta, \tag{9}$$

where $v_{\text{gain}}(x)$ and $v_{\text{loss}}(x)$ denote the positive and negative components of the perceived outcome, respectively. The

*Figure 3.* From rational to behaviorally realistic modeling: Pro-SF reformulates agent realistic behavior and provides robust outcomes.

parameters $\alpha, \beta \in (0, 1]$ encode diminishing sensitivity, while $\kappa > 1$ amplifies the perceived weight of losses.

This value function introduces an explicit asymmetry between gains and losses.

> For example, when $\kappa = 1.25$, a perceived gain of 9 and a perceived loss of 8 are evaluated asymmetrically as 9 versus 10 (i.e., $\kappa \cdot 8 = 10$), causing the loss component to dominate the decision.

**Probability distortion.** Agents often distort probabilities rather than evaluating them objectively, overweighting small chances of success and underweighting near certainties.

Let $p \in [0, 1]$ denote the objective probability of acceptance. The agent's subjective probability $w(p)$ is modeled via the probability weighting function:

$$w(p) = \frac{p^\gamma}{\left(p^\gamma + (1-p)^\gamma\right)^{1/\gamma}}, \quad \gamma \in (0, 1], \quad (10)$$

where $\gamma$ controls the curvature: smaller values yield a more pronounced inverse-$S$ shape, leading to stronger overweighting of small probabilities and underweighting of large ones.

> For example, with $\gamma = 0.6$, a small objective probability $p = 0.05$ is overweighted (i.e., $w(p) = 0.2 > p$), while a large objective probability $p = 0.8$ is underweighted (i.e., $w(p) = 0.6 < p$).

**Remark 1.** Eq. (10) follows standard probability-weighting functions used in prospect theory, which produce an inverse-$S$ shape probability weighting (Kahneman & Tversky, 2013; Barberis et al., 2016; Gonzalez & Wu, 1999).

**Reference bias.** Without loss of generality, let $s(x) \in [0, 1]$ denote the system-defined likelihood of acceptance, where $s(x) = 1$ corresponds to meeting the passing threshold. Rather than evaluating $s(x)$ precisely, agents often form a coarse-grained subjective reference point $r$ that reflects limited sensitivity in self-assessment. We model this reference point via a quantization operator:

$$r = \frac{1}{K} \left\lfloor K s(x) \right\rfloor, \quad (11)$$

where $K$ sets the step size $1/K$ (yielding $K+1$ discrete

levels), mapping $r$ onto equally spaced discrete levels (e.g., $K = 5$ gives $r \in \{0, 0.2, 0.4, 0.6, 0.8, 1.0\}$).

> For example, a student after an exam may only estimate their score within a rough range (e.g., "around 70–80") rather than a precise value (e.g., 73).

**Unified Prospect-Based Utility.** We now unify the three behavioral components into a single prospect-based utility that governs agents' manipulation choices.

**Definition 5.2** (Prospect-based utility in strategic manipulation). *For an agent with subjective reference point $r \in [0, 1]$, the prospect-based utility $U_P$ for manipulating features from $x$ to $x'$ is*

$$U_P(x, x'; \phi) = w^+\big(p(x')\big)\big(1 - r\big)^\alpha \\ - \kappa\, w^-\big(1 - p(x')\big)\, r^\beta - \kappa \cdot \big(\lambda c(x, x')^\beta\big), \quad (12)$$

*where $p(x') \in [0, 1]$ denotes the* model-implied *probability of being classified as positive after manipulation, and $\phi = \{\alpha, \beta, \kappa, \gamma\}$ denotes the Prospect-Theoretic parameter set.*

*The parameters $\alpha, \beta \in (0, 1]$ capture diminishing sensitivity, $\kappa > 1$ encodes loss aversion, $\lambda > 0$ controls the strength of manipulation cost, and $w^+, w^-$ are probability-weighting functions that distort objective probabilities into subjective decision weights.*

**Remark 2.** We treat the manipulation cost $c(x, x')$ as an effort expenditure that contributes to the agent's subjective loss, consistent with the loss-aversion principle in prospect theory (Passarelli & Del Ponte, 2020). In special cases where effort is treated as an objective friction rather than a psychologically salient loss, the cost term can be simplified to $-\lambda\, c(x, x')^\beta$.

As illustrated in Fig. 3, Pro-SF reformulates agents' strategic manipulation by jointly accounting for loss aversion, reference dependence, and probability distortion.

**Remark 3** (Information structure of agents). In our setting, agents do not have direct access to the current classifier $f_\theta$. Instead, they observe historical outcomes and use past feedback to form a subjective belief about their likelihood of acceptance, captured by $p(x') \in [0, 1]$ in Eq. (12). This belief is further distorted by psychological biases rather than computed directly from $f_\theta$, positioning our setting between

---

**Algorithm 1** Prospect-Guided Strategic Framework

---

**Require:** Dataset $\mathcal{D} = \{(x_i, y_i)\}_{i=1}^n$; a classifier $f \in \mathcal{F}$; manipulation cost $c(\cdot)$

1: **Parameters**: $\alpha, \beta \in (0,1]$, $\kappa > 1$, $\gamma \in (0,1]$, $K \in \mathbb{N}$
2: Initialize classifier $f \in \mathcal{F}$
3: **Perceive agents' manipulation with prospect-guided utility (Eq. (12)):**
4:      Anticipate *loss aversion* with Eq. (9)
5:      Anticipate *probability distortion* with Eq. (10)
6:      Anticipate *reference bias* with Eq. (11)
7: **Train classifier against manipulated data:**
8:      Construct $\tilde{\mathcal{D}} = \{(b_{\mathrm{P}}(x_i), y_i)\}_{i=1}^n$ with Eq. (13)
9:      Learn $f^* \in \arg\min_{f \in \mathcal{F}} \mathbb{E}_{(x,y)\sim\mathcal{D}} \big[ \mathcal{L}(f(b_{\mathrm{P}}(\mathbf{x})), y) \big]$
10: **Return** $f^\star$

---

the full-information assumption (Hardt et al., 2016) and a no-information extreme.

### 5.3. Integrating Pro-SF into Strategic Classification

We finally specify the strategic-response model and the learning objective under Pro-SF. The whole process of Pro-SF is illustrated in Algm. 1.

**Definition 5.3** (Prospect-guided Strategic Manipulation). *Given a classifier $f \in \mathcal{F}$ and data distribution $\mathcal{D}$, an agent strategically manipulates features in response to $f$:*

$$b_{\mathrm{P}}(\mathbf{x}) \in \arg\max_{\mathbf{x}' \in \mathcal{X}} U_{\mathrm{P}}(\mathbf{x}, \mathbf{x}'; \phi), \qquad (13)$$

*where $U_{\mathrm{P}}(\cdot; \phi)$ is the prospect-based utility defined in Eq. (12).*

**Definition 5.4** (Learning objective of Pro-SF). *The decision maker trains a classifier $f$ that minimizes the expected classification loss under prospect-guided manipulation:*

$$f^* \in \arg\min_{f \in \mathcal{F}} \mathbb{E}_{(x,y)\sim\mathcal{D}} \big[ \mathcal{L}(f(b_{\mathrm{P}}(\mathbf{x})), y) \big], \qquad (14)$$

*where $\mathcal{L}$ is a loss function for classification.*

### 5.4. Equilibrium Analysis

The Stackelberg game induced by Pro-SF inherits well-defined game-theoretic properties. Specifically, we establish three structural guarantees for the Pro-SF game:

- A Stackelberg equilibrium always exists under standard compactness conditions on the action space and classifier parameter space.

- The equilibrium is locally stable, i.e., the best-response dynamics between the classifier and agents converge to the equilibrium from any sufficiently close initialization.

- The equilibrium is locally unique within its neighborhood of attraction.

Formal statements and proofs are provided in Appendix F.

**Learnability of behavioral parameters.** The Prospect-Theoretic parameter set $\phi = \{\alpha, \beta, \kappa, \gamma\}$ used in Pro-SF is inferred from observed manipulation behavior pairs $\{x_i, x_i'\}_{i=1}^n$ via maximum likelihood:

$$\phi^* = \arg\max_\phi \sum_i \log P_\phi(x_i' \mid x_i), \qquad (15)$$

where $P_\phi(x_i' \mid x_i)$ is modeled using a discrete-choice (softmax) likelihood over a finite candidate set of feasible manipulations (see details in Appendix I).

## 6. Experiment

### 6.1. Experimental Setup

**Dataset.** We evaluate our framework on five datasets, including four real-world and one synthetic benchmarks: *Credit*, *Adult*, *Diabetes*, *German*, *Spam*, and *Synthetic* (see detailed description in Appendix G.1).

**Agent behavior paradigms.** To evaluate robustness under different behavioral assumptions, we consider three strategic manipulations:

- **Fully rational.** The classical strategic classification paradigm, where agents manipulate their features to maximize utility in Eq. (1) (Hardt et al., 2016).

- **Non-rational.** Agents stochastically deviate from the rational best-response strategy, driven by behavioral mechanisms in behavioral economics (Kahneman & Tversky, 2013; Kuvcak et al., 2018; Leoneti & Gomes, 2021).

- **Mixed behavioral.** A heterogeneous population in which a proportion $\pi$ of agents behave fully rationally, while the remaining $(1 - \pi)$ exhibit non-rational strategic behavior.

**Metric.** We use **accuracy** as the primary metric and introduce two additional metrics: **over-defense error** *(ODE)* and **under-defense error** *(UDE)*. ODE measures false rejections caused by the classifier being overly defensive. UDE measures false acceptances due to non-rational manipulation of agents and insufficient defense of the classifier.

**Baseline.** We mainly conduct experiments with linear models, consistent with previous standard approaches (Chen et al., 2023; Shavit et al., 2020; Ghalme et al., 2021) with Mahalanobis distance for manipulation cost (Gavish et al., 2022).

**Implementation.** We implement the classifier $f$ as a logistic regression model trained with cross-entropy loss and learning rate $10^{-3}$. In prospect-theoretic utility, we set the curvature parameters $\alpha = 0.8$ and $\beta = 0.7$, the loss aversion coefficient $\kappa = 2.25$, and the probability-weighting parameter $\gamma = 0.7$ by default. In the mixed agent behavior

*Table 1.* Performance (%) of rational-based models and Pro-SF models within different agent manipulation paradigms.

| Classifier | Manipulation | Datasets | | | | | |
|---|---|---|---|---|---|---|---|
| | | *Adult* | *Credit* | *Diabetes* | *German* | *Spam* | *Synthetic* |
| *Rational-based* | *Rational* | $78.53_{\pm0.87}$ | $76.12_{\pm1.34}$ | $70.25_{\pm1.52}$ | $74.31_{\pm1.41}$ | $83.87_{\pm1.21}$ | $81.62_{\pm1.28}$ |
| | *Non-rational* | $72.42_{\pm1.85}$ | $69.54_{\pm2.11}$ | $64.73_{\pm1.89}$ | $68.21_{\pm2.03}$ | $78.38_{\pm1.94}$ | $73.20_{\pm2.22}$ |
| | *Mixed Behavior* | $74.31_{\pm1.51}$ | $72.12_{\pm1.87}$ | $66.81_{\pm1.65}$ | $70.43_{\pm1.92}$ | $78.16_{\pm1.63}$ | $75.15_{\pm1.74}$ |
| ***Pro-SF (ours)*** | *Rational* | $75.31_{\pm0.93}$ | $77.02_{\pm1.25}$ | $71.23_{\pm1.43}$ | $75.18_{\pm1.38}$ | $82.92_{\pm1.18}$ | $80.41_{\pm1.12}$ |
| | *Non-rational* | $\mathbf{81.50}_{\pm1.23}$ | $\mathbf{82.41}_{\pm1.37}$ | $\mathbf{74.52}_{\pm1.26}$ | $\mathbf{79.35}_{\pm1.42}$ | $\mathbf{89.34}_{\pm1.27}$ | $\mathbf{85.20}_{\pm1.31}$ |
| | *Mixed Behavior* | $\mathbf{78.68}_{\pm1.77}$ | $\mathbf{79.13}_{\pm1.63}$ | $\mathbf{72.01}_{\pm1.58}$ | $\mathbf{76.24}_{\pm1.71}$ | $\mathbf{86.71}_{\pm1.52}$ | $\mathbf{82.34}_{\pm1.56}$ |

*Table 2.* Performance of our ablation study on behavioral mechanisms.

| Classifier | Behavioral Factors | | | Metrics | | |
|---|---|---|---|---|---|---|
| | *Refe.* | *Prob.* | *Los.* | *Accuracy (%)* ↑ | *Over-defense error (%)* ↓ | *Under-defense error (%)* ↓ |
| $f_{Pro-sf}$ | ✓ | ✓ | ✓ | $78.92_{\pm0.13}$ | $5.17_{\pm0.11}$ | $3.22_{\pm0.09}$ |
| $f_{Refe+Prob}$ | ✓ | ✓ | ✗ | $77.72_{\pm0.27}$ | $6.24_{\pm0.08}$ | $5.29_{\pm0.12}$ |
| $f_{Refe+Los}$ | ✓ | ✗ | ✓ | $77.80_{\pm0.16}$ | $7.21_{\pm0.10}$ | $5.05_{\pm0.07}$ |
| $f_{Prob+Los}$ | ✗ | ✓ | ✓ | $77.45_{\pm0.09}$ | $6.28_{\pm0.06}$ | $4.79_{\pm0.13}$ |
| $f_{Refe}$ | ✓ | ✗ | ✗ | $75.33_{\pm0.11}$ | $9.12_{\pm0.09}$ | $7.27_{\pm0.10}$ |
| $f_{Prob}$ | ✗ | ✓ | ✗ | $73.50_{\pm0.15}$ | $9.26_{\pm0.08}$ | $9.23_{\pm0.12}$ |
| $f_{Los}$ | ✗ | ✗ | ✓ | $75.91_{\pm0.11}$ | $7.14_{\pm0.11}$ | $7.05_{\pm0.09}$ |

*Note:* 1) *Refe.* = Reference bias. 2) *Prob.* = Probability distortion. 3) *Los.* = Loss aversion.

paradigm, the proportion is set as $\pi = 0.2$. More implementation details are included in Appendix G.3

## 6.2. Ablation Study

To better understand the design of our Pro-SF, we conduct the following ablation studies.

**Ablation on behavioral mechanisms.** We first examine the necessity of the three core behavioral mechanisms: loss aversion, probability weighting, and reference bias. Starting from the full model, we construct variants by *neutralizing* one component at a time. For example, the **rational-weighting** variant removes probability distortion by setting $w^+(p) = p$ and $w^-(1 - p) = 1 - p$.

**Parameter sensitivity.** We further assess robustness to parameter choices by varying one parameter (or parameter pair). Specifically, we consider:

- $(\alpha, \beta) \in \{(0.85, 0.8), (0.85, 0.75), (0.8, 0.8), (0.8, 0.7)\}$;
- $\kappa \in \{1.75, 2.0, 2.25, 2.5, 2.75\}$;
- $\gamma \in \{0.6, 0.65, 0.7, 0.8\}$,
- $r$ determined by the reference granularity $K \in \{3, 4, 5\}$.

A notation table is provided in Appendix G.2 ( Table 4) and more results are included in Appendix G.4(Fig. 6) [1] .

---

[1]The tested parameter ranges are determined based on our parameter learning objective (Eq. (15)) and ranges commonly adopted in prospect-theoretic and behavioral economics litera-

**An alternative of probability distortion modeling** is the two-parameter *Prelec* function (Prelec, 1998), $w(p) = \exp\left(-\eta(-\ln p)^\phi\right)$, $\phi, \eta > 0$. We examine this function in Appendix G.4 (Table 6).

## 6.3. Result and Analysis

**Overall performance.** As shown in Table 1, Pro-SF consistently outperforms rational-based models when agents exhibit behavioral deviations, both in the non-rational and the more realistic mixed settings, across all datasets. At the same time, Pro-SF maintains competitive performance under the fully rational paradigm, indicating that incorporating behavioral modeling does not compromise performance when classical assumptions hold. Overall, these results demonstrate that Pro-SF provides robust performance across diverse strategic environments.

**Ablation on behavioral mechanisms.** Table 2 shows that each component contributes meaningfully, and the full Pro-SF (all three enabled) consistently performs best. *Reference bias* yields the largest marginal gain: removing it hurts most because the model can no longer capture abstention, which is key to correcting over-defense. Dropping *probability weighting* reduces robustness to distorted tail probabilities and increases under-defense, while removing *loss aversion* eliminates gain–loss asymmetry and weakens boundary stability. Overall, multi-mechanism variants outperform single-

---

ture (Kahneman & Tversky, 2013; Borkar & Chandak, 2021).

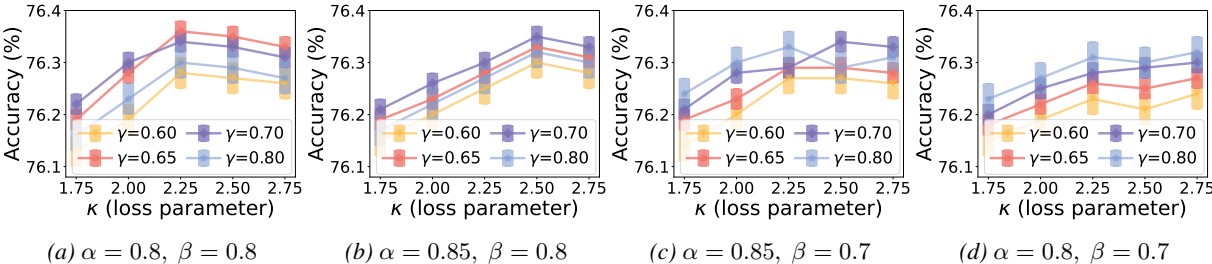

*Figure 4.* Parameter ablation results with parameters $(\alpha, \beta, \kappa, \gamma)$ and $r \in \{0, 0.2, 0.4, 0.6, 0.8\}$.

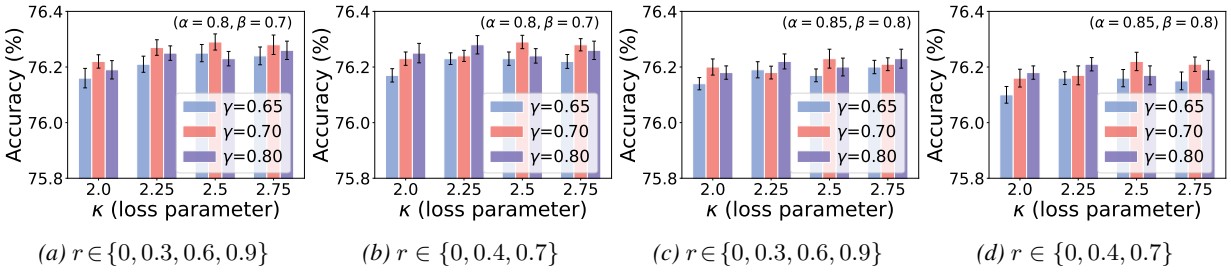

*Figure 5.* Parameter ablation results with parameters $\kappa, \gamma, r$ with different $(\alpha, \beta)$.

mechanism ones, indicating strong complementarity: reference bias governs *whether* agents move, whereas probability weighting and loss aversion determine *how far* they move.

**Parameter sensitivity.** Fig. 4 and Fig. 5 examine the effect of varying the Pro-SF parameters. Overall, accuracy remains stable across wide ranges, showing that Pro-SF does not rely on fine-tuned hyperparameters. Specifically, adjusting the loss aversion coefficient $\kappa$ only changes outcomes marginally, suggesting robustness to different levels of risk sensitivity. Variations in the probability distortion parameter $\gamma$ shift the relative emphasis on tail events but do not alter the overall trend. Different bins of $r$ only cause some fluctuations in performance, but are all better than the rational classifier. Finally, different curvature settings $(\alpha, \beta)$ yield consistent results, confirming that Pro-SF maintains effectiveness under diverse utility shapes.

## 7. Conclusion

This work challenges the classical rational-agent assumption in strategic classification and formalizes the problem of *behaviorally realistic strategic classification*. We theoretically characterize the impact of behavioral mismatch on deployment performance, and propose the *Prospect-Guided Strategic Framework*, which incorporates three key psychological mechanisms underlying strategic manipulation and provides a behaviorally grounded solution for SC in the real-world. Future work will extend this framework to richer and more diverse deployment settings.

## Acknowledgement

This work was jointly supported by the National Natural Science Foundation of China (Nos. 62276004, 62325604, 62525213), the Beijing Natural Science Foundation (No. L257007), the Beijing Major Science and Technology Project (No. Z251100008425006), the Shanghai Sailing Program (No. 24YF2711600), the Provincial Natural Science Foundation of Heilongjiang Province (No. LH2023C069), the Provincial Natural Science Foundation of Hunan Province (No. 2025JJ10008), the NUDT Youth Independent Innovation Science Fund (No. ZK25-20), the NUDT Innovation Foundation for Postgraduate (No. XJQY2025011), and the State Key Laboratory of General Artificial Intelligence.

## Impact Statement

This paper presents work whose goal is to advance the field of Machine Learning. There are many potential societal consequences of our work, none of which we feel must be specifically highlighted here.

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

## Appendix Contents

## A. Additional Related Work

### A.1. More Strategic Machine Learning Work

Several additional directions in strategic machine learning complement the studies highlighted in Section 2.

**Robustness extensions.** Beyond classical defenses, researchers have explored stochastic classifiers (Singh & Kulkarni, 2024), differentiable optimization layers for end-to-end robustness (Levanon & Rosenfeld, 2021), and graph-based models to capture inter-agent dependencies (Eilat et al., 2023). Multi-agent formulations further investigate externalities and collective dynamics in strategic settings (Hossain et al., 2025; Kou et al., 2025; 2026).

**Positive manipulation.** A growing body of work highlights the constructive potential of strategic behavior. For example, classifiers that incentivize authentic qualification gains have been proposed in education and hiring contexts (Kleinberg & Raghavan, 2020; Harris et al., 2022; Liu et al., 2026). These approaches complement causal frameworks (Miller et al., 2020; Chen et al., 2023; 2026), which distinguish between manipulable and improvable features.

**Performative prediction extensions.** Beyond the foundational contributions (Perdomo et al., 2020; Rosenfeld et al., 2020; Hardt et al., 2022; Mendler-Dünner et al., 2022), further work explores neural methods for dynamic feedback loops (Mofakhami et al., 2023) and more recent advances in model-induced distribution shifts (Hardt & Mendler-Dünner, 2025).

**Fairness perspectives.** In addition to direct fairness interventions (Zhang et al., 2022; Estornell et al., 2023b; Keswani & Celis, 2023; Lv et al., 2026c), several studies consider fairness as an incentive-alignment mechanism that influences agents' strategic behavior, thereby connecting individual manipulation with broader social equity.

### A.2. Position of Our Work

Our work, Pro-SF, is designed to *complement* the classical strategic classification paradigm rather than overturn it. The rational-agent model remains a useful idealization—and a natural baseline—for analyzing strategic behavior under well-specified utilities and costs. Our contribution is to provide a behaviorally grounded formulation for settings where strategic responses systematically deviate from this idealization due to psychological biases. Importantly, Pro-SF is fully compatible with classical SC: when behavioral mechanisms are absent (e.g., $\kappa = 1$ and probability weighting reduces to the identity), Pro-SF reduces to the standard rational-response formulation.

### A.3. Additional Behavioral Economics Background

Beyond the works highlighted in Section 2, a broader body of literature illustrates how behavioral economics explains systematic deviations from rationality and informs computational models.

**Bounded rationality and heuristics.** Human decision-making is constrained by limited cognitive resources and often relies on simplified rules rather than exhaustive optimization (Todd & Gigerenzer, 2000; Evans & Prokopenko, 2024; Wang et al., 2023). This perspective suggests that agents may respond in partial, myopic, or context-dependent ways instead of precise best responses.

**Temporal and framing effects.** Individuals overweight immediate costs relative to delayed benefits and respond differently to equivalent options depending on presentation (Laibson, 1997; O'donoghue & Rabin, 1999). These effects imply that manipulations which appear objectively beneficial may still be avoided or inconsistently adopted.

**Extended applications.** Behavioral frameworks have been applied across diverse algorithmic domains. Examples include portfolio optimization and asset pricing in finance (Barberis et al., 2021), multi-criteria decision methods (Leoneti & Gomes, 2021), fairness-aware allocation in management science (Holmes Jr et al., 2011), and human–AI decision support systems (Payne, 2025). Collectively, these studies demonstrate the versatility of prospect-theoretic and behavioral models in explaining and predicting non-rational behavior.

# B. Proof of Proposition 4.1

### B.1. Assumptions and Setup

We work under a threshold classifier $f_R(x) = \mathbf{1}\{s(x) \geq \tau\}$, where $s : \mathcal{X} \to \mathbb{R}$ is the decision score and $\tau \in \mathbb{R}$ is the threshold, with the 0–1 loss $\mathcal{L}(f(x), y) = \mathbf{1}\{f(x) \neq y\}$. The deployment error reduces to:

$$\delta(\theta) = \mathbb{P}_{(x,y)\sim\mathcal{D}_\theta}\big(f_R(x) \neq y\big). \tag{16}$$

The key assumption is the *boundary-mass condition*: there exist constants $\epsilon > 0$ and $\gamma > 0$ such that

$$\mathbb{P}_{(x,y)\sim\mathcal{D}_\theta}\Big(|s(x) - \tau| \leq \gamma, \ \ y \neq \mathbf{1}\{s(x) \geq \tau\}\Big) \ \geq \ \epsilon. \tag{17}$$

This condition states that under the true behavioral distribution $\mathcal{D}_\theta$, a non-negligible fraction of deployment samples falls within a $\gamma$-neighborhood of the decision boundary and carries labels that disagree with the classifier's prediction.

### B.2. Discussion: Naturality of the Boundary-Mass Condition

The boundary-mass condition (17) is not an artificial regularity assumption; it is a direct and natural consequence of the behavioral mismatch between $\mathcal{D}_\theta$ and $\mathcal{D}_R$ in strategic classification settings.

**Why the condition holds in practice.** The classifier $f_R$ is trained on the rational post-manipulation distribution $\mathcal{D}_R$, and its decision boundary $\tau$ is calibrated to correctly separate the positive and negative classes under $\mathcal{D}_R$. When agents deviate from rational behavior—due to loss aversion, reference bias, or probability distortion—the actual deployment distribution $\mathcal{D}_\theta$ differs from $\mathcal{D}_R$. This shift systematically places a portion of the agent population near the decision boundary with incorrect labels under $f_R$, for two concrete reasons:

- **Over-defense**: agents subject to loss aversion or reference bias may abstain from manipulation even when rational agents would manipulate to cross $\tau$. These abstaining agents remain on the "wrong side" of the boundary that $f_R$ was trained to defend, contributing label-disagreeing mass near $\tau$.

- **Under-defense**: agents subject to probability distortion may overshoot the rational manipulation endpoint, landing on the other side of $\tau$ in a region where $f_R$ was not designed to classify correctly.

In both cases, the mass accumulates specifically near the decision boundary because the classifier $f_R$ was optimized to be accurate far from $\tau$; the boundary region is precisely where the mismatch between $\mathcal{D}_R$ and $\mathcal{D}_\theta$ is most consequential. The parameters $\epsilon$ and $\gamma$ quantify the severity of this mismatch: larger behavioral deviations produce larger $\epsilon$ and $\gamma$, as confirmed empirically in Fig. 2c.

**Relation to distribution shift.** Condition (17) is strictly weaker than requiring $\mathcal{D}_\theta \neq \mathcal{D}_R$ globally; it only requires the distributional difference to manifest in a decision-relevant region. This is precisely the regime that matters for classification performance: differences far from the boundary have no impact on accuracy, while differences near the boundary do.

### B.3. Proof

Define the boundary disagreement set:

$$\mathcal{A} := \Big\{(x, y) : |s(x) - \tau| \leq \gamma, \ \ y \neq \mathbf{1}\{s(x) \geq \tau\}\Big\}. \tag{18}$$

For any $(x, y) \in \mathcal{A}$, the classifier predicts $f_R(x) = \mathbf{1}\{s(x) \geq \tau\} \neq y$, so $\mathcal{L}(f_R(x), y) = 1$. Therefore:

$$\delta(\theta) = \mathbb{E}_{(x,y)\sim\mathcal{D}_\theta}\big[\mathbf{1}\{f_R(x) \neq y\}\big] \tag{19}$$

$$\geq \mathbb{E}_{(x,y)\sim\mathcal{D}_\theta}\big[\mathbf{1}\{(x, y) \in \mathcal{A}\}\big] \tag{20}$$

$$= \mathbb{P}_{(x,y)\sim\mathcal{D}_\theta}\big((x, y) \in \mathcal{A}\big) \tag{21}$$

$$\geq \epsilon, \tag{22}$$

where the last inequality uses the boundary-mass condition (17). Hence $\delta(\theta) \geq \epsilon > 0$, establishing that the deployment error is strictly bounded away from zero and cannot be eliminated by optimizing under the rational-agent assumption.

## C. Proof of Proposition 4.3

### C.1. Assumptions and Setup

Recall that $b_R(x)$ denotes the rational best response (Eq. (1)), and $f_R$ is the classifier optimized under the rational-agent assumption, i.e., an empirical risk minimizer of

$$R_{\text{train}}(f) = \mathbb{E}\big[\mathcal{L}(f(b_R(X)), Y)\big] \quad \text{(cf. Eq. (2)).} \tag{23}$$

Intuitively, $f_R$ is trained as if every agent manipulates to the rational endpoint $b_R(x)$. Under behavioral biases, however, some agents abstain from manipulation entirely. This training–deployment mismatch is the root cause of over-defense.

We define the *abstention set*:

$$A = \big\{x : b_R(x) \neq x \ \text{and the agent actually stays at } x\big\}, \tag{24}$$

i.e., inputs where the rational model predicts manipulation but the behavioral agent abstains. We assume $P(A) > 0$, so abstention occurs with non-negligible probability.

On the abstention set $A$, we define two key quantities:

$$\tau_A := \Pr\big(f_R(X) \neq f_R(b_R(X)) \mid X \in A\big), \tag{25}$$

$$\varepsilon_R := \Pr\big(f_R(b_R(X)) \neq Y \mid X \in A\big). \tag{26}$$

Here $\tau_A$ measures the probability that $f_R$ predicts differently at the true feature $x$ versus the rational endpoint $b_R(x)$—capturing the extent to which the decision boundary has shifted due to over-defense. The quantity $\varepsilon_R$ is the residual training error of $f_R$ at the rational endpoints $b_R(x)$, which is small when the classifier fits the training distribution well.

### C.2. Discussion: Naturality of the Assumptions

**Why $P(A) > 0$ holds in practice.** The abstention set $A$ is non-empty whenever behavioral biases cause agents to forgo manipulation that rational agents would undertake. This is precisely the over-defense scenario: agents subject to loss aversion perceive the manipulation cost as disproportionately large, and agents subject to reference bias may assess themselves as too far from the acceptance threshold to bother trying. Both mechanisms are well-documented in behavioral economics (Kahneman & Tversky, 2013) and are directly captured by Pro-SF. The assumption $P(A) > 0$ therefore holds whenever any fraction of the agent population exhibits these biases, which is the defining condition of the BR-SC problem.

**Why $\tau_A > 2\varepsilon_R$ holds in practice.** The classifier $f_R$ is trained on the post-manipulation distribution $\mathcal{D}_R$, so its decision boundary $\tau$ is optimized to correctly classify agents at their rational endpoints $b_R(x)$. Consequently, $\varepsilon_R$ is small: $f_R$ performs well at $b_R(x)$ by construction. In contrast, $\tau_A$ can be large: since $f_R$ shifts its boundary outward to defend against rational manipulations, it systematically misclassifies agents who stay at $x \in A$ and are on the "wrong side" of the shifted boundary. The condition $\tau_A > 2\varepsilon_R$ thus formalizes the intuition that over-defense hurts most when the boundary shift is non-trivial and the classifier is well-trained—a regime that is typical in practice, as confirmed by the monotone increase of over-defense error with behavioral deviation in Fig. 2c.

## C.3. Proof

**Step 1: Loss decomposition on** $A$**.** For any classifier $f$ and any $(x, y)$ with $x \in A$, the difference in 0–1 loss between the true feature $x$ and the rational endpoint $b_R(x)$ satisfies:

$$\mathcal{L}(f(x), y) - \mathcal{L}(f(b_R(x)), y) = \mathbf{1}\{f(x) \neq y, \, f(b_R(x)) = y\} - \mathbf{1}\{f(x) = y, \, f(b_R(x)) \neq y\}. \tag{27}$$

Taking conditional expectation over $X \in A$:

$$\mathbb{E}\big[\mathcal{L}(f(X), Y) - \mathcal{L}(f(b_R(X)), Y) \mid X \in A\big]$$
$$= \Pr(f(X) \neq Y, \, f(b_R(X)) = Y \mid X \in A) - \Pr(f(X) = Y, \, f(b_R(X)) \neq Y \mid X \in A). \tag{28}$$

**Step 2: Lower bound via set inclusions.** We apply the following two inclusions:

$$\{f(X) \neq Y, \, f(b_R(X)) = Y\} \supseteq \{f(X) \neq f(b_R(X))\} \setminus \{f(b_R(X)) \neq Y\}, \tag{29}$$

$$\{f(X) = Y, \, f(b_R(X)) \neq Y\} \subseteq \{f(b_R(X)) \neq Y\}. \tag{30}$$

Inclusion (29) states that whenever $f$ disagrees between $x$ and $b_R(x)$, this contributes to the first term unless $b_R(x)$ is already mislabeled. Inclusion (30) states that the second term is bounded by the training error at $b_R(x)$. Substituting into (28) gives the lower bound:

$$\mathbb{E}\big[\mathcal{L}(f(X), Y) - \mathcal{L}(f(b_R(X)), Y) \mid X \in A\big] \geq \Pr(f(X) \neq f(b_R(X)) \mid X \in A) - 2\Pr(f(b_R(X)) \neq Y \mid X \in A). \tag{31}$$

**Step 3: Apply to** $f_R$ **and globalize.** Substituting $f = f_R$ into (31) and using the definitions of $\tau_A$ and $\varepsilon_R$ in Eqs. (25)–(26):

$$\mathbb{E}\big[\mathcal{L}(f_R(X), Y) - \mathcal{L}(f_R(b_R(X)), Y) \mid X \in A\big] \geq \tau_A - 2\varepsilon_R. \tag{32}$$

Multiplying by $P(A) > 0$ and adding the contribution from $X \notin A$:

$$\mathbb{E}[\mathcal{L}(f_R(X), Y)] - \mathbb{E}[\mathcal{L}(f_R(b_R(X)), Y)] \geq P(A)(\tau_A - 2\varepsilon_R). \tag{33}$$

Under the condition $\tau_A > 2\varepsilon_R$, the right-hand side is strictly positive, yielding:

$$\mathbb{E}[\mathcal{L}(f_R(x), y)] > \mathbb{E}[\mathcal{L}(f_R(b_R(x)), y)]. \tag{34}$$

**Remark 4.** Geometrically, $f_R$ shifts its decision boundary outward to guard against the rational endpoints $b_R(x)$. On the abstention set $A$, agents remain at $x$, which now falls on the wrong side of the shifted boundary. The quantity $\tau_A - 2\varepsilon_R$ measures the net excess deployment error due to this over-defense: $\tau_A$ count the fraction of abstaining agents misclassified by the shifted boundary, while $2\varepsilon_R$ accounts for the small residual error $f_R$ already makes at the rational endpoints it was trained on.

# D. Proof of Proposition 4.5

### D.1. Assumptions and Setup

Recall that $b_R(x)$ denotes the rational best response (Eq. (1)), and $f_R$ is the classifier optimized under the rational-agent assumption, i.e., an empirical risk minimizer of

$$R_{\text{train}}(f) = \mathbb{E}\big[\mathcal{L}(f(b_R(X)), Y)\big] \quad \text{(cf. Eq. (2)).} \tag{35}$$

Intuitively, $f_R$ is trained to defend against manipulations that end exactly at $b_R(x)$. Under behavioral biases such as probability distortion and loss aversion, however, agents may overshoot the rational endpoint and land at $b_\theta(x)$ with $b_\theta(x) \neq b_R(x)$. Since $f_R$ was never trained to handle these overshoot points, its defense is insufficient—the essence of under-defense.

We define the *overshoot set*:

$$B = \big\{x : b_R(x) \neq x \text{ and } b_\theta(x) \neq b_R(x) \text{ with overshoot beyond } b_R(x)\big\}, \tag{36}$$

i.e., inputs where the agent manipulates further than the rational endpoint along the manipulation direction. We assume $P(B) > 0$, so overshoot occurs with non-negligible probability.

On the overshoot set $B$, we define two key quantities:

$$\tau_B := \Pr\big(f_R(b_\theta(X)) \neq f_R(b_R(X)) \mid X \in B\big), \tag{37}$$

$$\varepsilon_R := \Pr\big(f_R(b_R(X)) \neq Y \mid X \in B\big). \tag{38}$$

Here $\tau_B$ measures the probability that $f_R$ predicts differently at the overshoot point $b_\theta(x)$ versus the rational endpoint $b_R(x)$—capturing how often agents land in regions where $f_R$'s defense fails. The quantity $\varepsilon_R$ is the residual training error of $f_R$ at the rational endpoints, which is small when the classifier fits the training distribution well.

### D.2. Discussion: Naturality of the Assumptions

**Why $P(B) > 0$ holds in practice.** The overshoot set $B$ is non-empty whenever behavioral biases drive agents beyond the rational manipulation endpoint. Two mechanisms in Pro-SF directly produce this:

- **Probability distortion**: agents overweight small probabilities of acceptance, causing them to perceive a higher chance of success than actually exists. This inflated belief motivates more aggressive manipulation, pushing agents past the rational endpoint $b_R(x)$.

- **Loss aversion**: the fear of rejection amplifies the perceived cost of falling just short of the acceptance threshold, leading agents to overshoot as a precautionary measure to ensure they clear the boundary.

Both mechanisms are integral to the BR-SC problem setting, so $P(B) > 0$ holds whenever any fraction of the population exhibits these biases—which is the defining condition of the problem we study.

**Why $\tau_B > 2\varepsilon_R$ holds in practice.** The classifier $f_R$ is trained on $\mathcal{D}_R$, so it is well-calibrated at the rational endpoints $b_R(x)$: the residual error $\varepsilon_R$ is small by construction. In contrast, $\tau_B$ can be large because the overshoot region $b_\theta(x)$ lies outside the support of the training distribution $\mathcal{D}_R$. The classifier $f_R$ has no training signal in this region, so its predictions there are unreliable. The condition $\tau_B > 2\varepsilon_R$. Therefore, captures the typical regime where the classifier performs well on its training distribution but poorly on the unseen overshoot region—a standard manifestation of distribution shift, confirmed empirically by the monotone increase of under-defense error with behavioral deviation in Fig. 2c.

### D.3. Proof

**Step 1: Loss decomposition on $B$.** For any classifier $f$ and any $(x, y)$ with $x \in B$, the difference in 0–1 loss between the overshoot point $b_\theta(x)$ and the rational endpoint $b_R(x)$ satisfies:

$$\mathcal{L}(f(b_\theta(x)), y) - \mathcal{L}(f(b_R(x)), y) = \mathbf{1}\{f(b_\theta(x)) \neq y, \, f(b_R(x)) = y\} - \mathbf{1}\{f(b_\theta(x)) = y, \, f(b_R(x)) \neq y\}. \tag{39}$$

Taking conditional expectation over $X \in B$:

$$\mathbb{E}\big[\mathcal{L}(f(b_\theta(X)), Y) - \mathcal{L}(f(b_R(X)), Y) \mid X \in B\big]$$
$$= \Pr(f(b_\theta(X)) \neq Y, \, f(b_R(X)) = Y \mid X \in B) - \Pr(f(b_\theta(X)) = Y, \, f(b_R(X)) \neq Y \mid X \in B). \tag{40}$$

**Step 2: Lower bound via set inclusions.** We apply the following two inclusions:

$$\{f(b_\theta) \neq Y, \, f(b_R) = Y\} \supseteq \{f(b_\theta) \neq f(b_R)\} \setminus \{f(b_R) \neq Y\}, \tag{41}$$

$$\{f(b_\theta) = Y, \, f(b_R) \neq Y\} \subseteq \{f(b_R) \neq Y\}. \tag{42}$$

Inclusion (41) states that whenever $f$ disagrees between $b_\theta(x)$ and $b_R(x)$, this contributes to the loss increase unless $b_R(x)$ is already mislabeled. Inclusion (42) states that the loss-decreasing term is bounded by the training error at $b_R(x)$. Substituting into (40) gives the lower bound:

$$\mathbb{E}\big[\mathcal{L}(f(b_\theta(X)), Y) - \mathcal{L}(f(b_R(X)), Y) \mid X \in B\big] \geq \Pr(f(b_\theta(X)) \neq f(b_R(X)) \mid X \in B) - 2\Pr(f(b_R(X)) \neq Y \mid X \in B). \tag{43}$$

**Step 3: Apply to $f_R$ and globalize.** Substituting $f = f_R$ into (43) and using the definitions of $\tau_B$ and $\varepsilon_R$ in Eqs. (37)–(38):

$$\mathbb{E}\big[\mathcal{L}(f_R(b_\theta(X)), Y) - \mathcal{L}(f_R(b_R(X)), Y) \mid X \in B\big] \geq \tau_B - 2\,\varepsilon_R. \tag{44}$$

Multiplying by $P(B) > 0$ and adding the contribution from $X \notin B$:

$$\mathbb{E}[\mathcal{L}(f_R(b_\theta(X)), Y)] - \mathbb{E}[\mathcal{L}(f_R(b_R(X)), Y)] \geq P(B)\,(\tau_B - 2\,\varepsilon_R). \tag{45}$$

Under the condition $\tau_B > 2\varepsilon_R$, the right-hand side is strictly positive, yielding:

$$\mathbb{E}\big[\mathcal{L}(f_R(b_\theta(x)), y)\big] > \mathbb{E}\big[\mathcal{L}(f_R(b_R(x)), y)\big]. \tag{46}$$

$\square$

**Remark 5.** Geometrically, $f_R$ learns to guard the rational endpoints $b_R(x)$, but the overshoot region around $b_\theta(x)$ lies outside the support of its training distribution. The quantity $\tau_B - 2\varepsilon_R$ measures the net excess deployment error: $\tau_B$ counts the fraction of overshooting agents that land where $f_R$ makes a different and incorrect prediction, while $2\varepsilon_R$ discounts the small residual error $f_R$ already makes at the rational endpoints it was trained on.

## E. Pro-SF Improves Deployment Guarantees

In this appendix, we show that, by reducing the mismatch between the training-time post-manipulation distribution and the deployment distribution, Pro-SF yields a strictly stronger deployment-performance guarantee than training under the rational-agent assumption.

### E.1. Setup.

Let $\mathcal{D}_\theta$ denote the true post-manipulation (deployment) distribution induced by the behavioral response $b_\theta$. Let $\mathcal{D}_R$ denote the post-manipulation distribution induced by the rational response $b_R$. Similarly, let $\mathcal{D}_P$ denote the post-manipulation distribution induced by the Prospect-Guided response $b_P$ (Pro-SF). Define the population risk under a distribution $\mathcal{D}$ by

$$R_\mathcal{D}(f) := \mathbb{E}_{(x,y)\sim\mathcal{D}}\big[\mathcal{L}(f(x), y)\big], \tag{47}$$

where the loss satisfies $0 \leq \mathcal{L} \leq 1$. Let

$$f_R \in \arg\min_f R_{\mathcal{D}_R}(f), \qquad f_P \in \arg\min_f R_{\mathcal{D}_P}(f) \tag{48}$$

be classifiers optimized on the rational and Pro-SF induced training distributions, respectively. Finally, let

$$R^\star := \inf_f R_{\mathcal{D}_\theta}(f) \tag{49}$$

denote the Bayes-optimal (best achievable) deployment risk under the true post-manipulation distribution $\mathcal{D}_\theta$.

### E.2. Lemma (TV controls expectation shift).

For any two distributions $P, Q$ on $\mathcal{X} \times \mathcal{Y}$ and any measurable function $g : \mathcal{X} \times \mathcal{Y} \to [0, 1]$,

$$\big|\mathbb{E}_P[g] - \mathbb{E}_Q[g]\big| \leq \mathrm{TV}(P, Q). \tag{50}$$

*Proof.* This is a standard property of total variation distance: by the variational characterization, $\mathrm{TV}(P, Q) = \sup_A |P(A) - Q(A)| = \sup_{0 \leq g \leq 1} |\mathbb{E}_P[g] - \mathbb{E}_Q[g]|$. Applying it to the given $g \in [0, 1]$ yields (50). $\square$

**Proposition (excess deployment risk is controlled by mismatch).** For any surrogate training distribution $\widehat{\mathcal{D}}$ and the corresponding optimizer $\widehat{f} \in \arg\min_f R_{\widehat{\mathcal{D}}}(f)$, the deployment excess risk satisfies

$$R_{\mathcal{D}_\theta}(\widehat{f}) - R^\star \leq 2\,\mathrm{TV}\big(\mathcal{D}_\theta, \widehat{\mathcal{D}}\big). \tag{51}$$

*Proof.* Let $f^\star \in \arg\min_f R_{\mathcal{D}_\theta}(f)$ be a Bayes-optimal classifier under $\mathcal{D}_\theta$. By Lemma (50) applied to the bounded loss, for any classifier $f$,

$$R_{\mathcal{D}_\theta}(f) \leq R_{\widehat{\mathcal{D}}}(f) + \mathrm{TV}(\mathcal{D}_\theta, \widehat{\mathcal{D}}), \qquad R_{\widehat{\mathcal{D}}}(f) \leq R_{\mathcal{D}_\theta}(f) + \mathrm{TV}(\mathcal{D}_\theta, \widehat{\mathcal{D}}). \tag{52}$$

Using $\widehat{f}$ optimal for $\widehat{\mathcal{D}}$,

$$R_{\widehat{\mathcal{D}}}(\widehat{f}) \leq R_{\widehat{\mathcal{D}}}(f^\star). \tag{53}$$

Now chain the inequalities:

$$\begin{aligned}
R_{\mathcal{D}_\theta}(\widehat{f}) &\leq R_{\widehat{\mathcal{D}}}(\widehat{f}) + \mathrm{TV}(\mathcal{D}_\theta, \widehat{\mathcal{D}}) \\
&\leq R_{\widehat{\mathcal{D}}}(f^\star) + \mathrm{TV}(\mathcal{D}_\theta, \widehat{\mathcal{D}}) \\
&\leq R_{\mathcal{D}_\theta}(f^\star) + 2\,\mathrm{TV}(\mathcal{D}_\theta, \widehat{\mathcal{D}}) \qquad \text{(by (52))} \\
&= R^\star + 2\,\mathrm{TV}(\mathcal{D}_\theta, \widehat{\mathcal{D}}).
\end{aligned}$$

Rearranging gives (51). □

### E.3. Corollary.

Applying (51) with $\widehat{\mathcal{D}} = \mathcal{D}_R$ and $\widehat{\mathcal{D}} = \mathcal{D}_P$ yields

$$R_{\mathcal{D}_\theta}(f_R) - R^\star \;\leq\; 2\,\mathrm{TV}(\mathcal{D}_\theta, \mathcal{D}_R), \tag{54}$$

and

$$R_{\mathcal{D}_\theta}(f_P) - R^\star \;\leq\; 2\,\mathrm{TV}(\mathcal{D}_\theta, \mathcal{D}_P). \tag{55}$$

Therefore, whenever Pro-SF reduces behavioral mismatch in the sense that

$$\mathrm{TV}(\mathcal{D}_\theta, \mathcal{D}_P) < \mathrm{TV}(\mathcal{D}_\theta, \mathcal{D}_R), \tag{56}$$

it provides a *strictly stronger* deployment-performance guarantee than training under the rational-agent assumption, since the right-hand side upper bound in (55) is strictly smaller than that in (54).

## F. Existence of Stackelberg Equilibrium Under Pro-SF

This appendix establishes three structural properties of the Stackelberg game induced by Pro-SF: existence of a Stackelberg equilibrium (Proposition F.2), local stability under best-response dynamics (Proposition F.3), and local uniqueness of the equilibrium (Corollary F.4).

### F.1. Setup

The Pro-SF game follows the classic Stackelberg structure with two players:

- **Leader** (classifier): chooses a parameter vector $\theta \in \Theta$, where $\Theta \subset \mathbb{R}^d$ is the classifier parameter space.

- **Follower** (agent): observes $\theta$ and chooses a manipulated feature vector $x' \in \mathcal{X}$ to maximize the prospect-based utility $U_P(x, x'; \phi)$ defined in Eq. (12).

Given any classifier parameter $\theta$, the agent's best-response function is:

$$b_P(x; \theta) \in \arg\max_{x' \in \mathcal{X}} U_P(x, x'; \phi), \tag{57}$$

and the leader's objective induced by this response is:

$$F(\theta) \;=\; \mathbb{E}_{(x,y) \sim \mathcal{D}}[\mathcal{L}(f_\theta(b_P(x; \theta)), y)]. \tag{58}$$

**Definition F.1** (Stackelberg Equilibrium under Pro-SF). *A pair $(\theta^*, b^*(\cdot))$ is a* Stackelberg equilibrium *of the Pro-SF game if:*

*(i) **Follower optimality**: for every $x \in \mathcal{X}$,*

$$b^*(x) \in \arg\max_{x' \in \mathcal{X}} U_P(x, x'; \phi); \tag{59}$$

*(ii) **Leader optimality**:*

$$\theta^* \in \arg\min_{\theta \in \Theta} \mathbb{E}_{(x,y) \sim \mathcal{D}}[\mathcal{L}(f_\theta(b^*(x)), y)]. \tag{60}$$

## F.2. Proposition: Existence of Stackelberg Equilibrium

**Proposition F.2** (Existence). *Under the Pro-SF framework, suppose the following conditions hold:*

*(i) The feasible manipulation space $\mathcal{X} \subset \mathbb{R}^p$ is compact;*

*(ii) The classifier parameter space $\Theta \subset \mathbb{R}^d$ is compact;*

*(iii) The prospect-based utility $U_P(x, x'; \phi)$ is continuous in $x'$ for every fixed $x$ and $\phi$;*

*(iv) The classifier score $f_\theta(x')$ is continuous in both $x'$ and $\theta$, and the loss function $\mathcal{L}$ is continuous.*

*Then the Pro-SF game admits at least one Stackelberg equilibrium $(\theta^*, b^*(\cdot))$.*

*Proof.* We establish existence in three steps.

**Step 1: Existence of the follower's best response.** Fix any $\theta \in \Theta$ and $x \in \mathcal{X}$. We verify that $U_P(x, x'; \phi)$ satisfies condition (iii) by inspecting each component in Eq. (12):

- The classifier score $p(x') = \sigma(f_\theta(x'))$ is continuous in $x'$ by condition (iv) and the continuity of the sigmoid $\sigma$;

- The value function $v(\cdot)$ in Eq. (9) is a power function with exponents $\alpha, \beta \in (0, 1]$, hence continuous on $\mathbb{R}_{\geq 0}$;

- The probability-weighting functions $w^+(\cdot)$ and $w^-(\cdot)$ in Eq. (10) are smooth for all $\gamma \in (0, 1]$;

- The reference point $r \in [0, 1]$ is a fixed scalar determined by Eq. (11), independent of $x'$;

- The manipulation cost $c(x, x')$ (e.g., Mahalanobis distance) is continuous in $x'$ for fixed $x$.

Since $U_P(x, x'; \phi)$ is a finite composition of continuous functions, it is continuous in $x'$. By condition (i), $\mathcal{X}$ is compact. Applying the Weierstrass extreme-value theorem, the maximum

$$\max_{x' \in \mathcal{X}} U_P(x, x'; \phi) \tag{61}$$

is attained, so the argmax set is non-empty:

$$b_P(x; \theta) := \arg\max_{x' \in \mathcal{X}} U_P(x, x'; \phi) \neq \varnothing. \tag{62}$$

Hence the follower's best-response function $b_P(x; \theta)$ exists for every $x \in \mathcal{X}$ and $\theta \in \Theta$.

**Step 2: Existence of the leader's optimum.** Given the follower's best-response mapping $b_P(\cdot; \theta)$ established in Step 1, the leader's objective is:

$$F(\theta) = \mathbb{E}_{(x,y) \sim \mathcal{D}}[\mathcal{L}(f_\theta(b_P(x; \theta)), y)]. \tag{63}$$

We show that $F$ is continuous on $\Theta$. By the maximum theorem (Berge, 1963), since $U_P(x, x'; \phi)$ is continuous in $(x', \theta)$ and $\mathcal{X}$ is compact, the correspondence $\theta \mapsto \arg\max_{x'} U_P(x, x'; \phi)$ is upper hemicontinuous, and the value function $\theta \mapsto \max_{x'} U_P(x, x'; \phi)$ is continuous in $\theta$. Under a measurable selection of $b_P(x; \theta)$, the composed map $\theta \mapsto f_\theta(b_P(x; \theta))$

is continuous in $\theta$ by condition (iv). Since $\mathcal{L}$ is continuous and the expectation preserves continuity (by the dominated convergence theorem, as $\mathcal{L}$ is bounded), $F(\theta)$ is continuous on $\Theta$. Since $\Theta$ is compact by condition (ii), the Weierstrass theorem guarantees the existence of

$$\theta^* \in \arg\min_{\theta \in \Theta} F(\theta). \tag{64}$$

**Step 3: Construction of the equilibrium.** Let $b^*(x) := b_P(x; \theta^*)$ for every $x \in \mathcal{X}$. By Step 1, $b^*(x) \in \arg\max_{x'} U_P(x, x'; \phi)$ for every $x$, so follower optimality holds. By Step 2, $\theta^*$ minimizes $F(\theta) = \mathbb{E}[\mathcal{L}(f_\theta(b^*(x)), y)]$ over $\Theta$, so leader optimality holds. Therefore, $(\theta^*, b^*(\cdot))$ is a Stackelberg equilibrium of the Pro-SF game. $\qquad\square$

**Remark 6.** Condition (i) is naturally satisfied in practice: real-world features (income, credit score, health metrics) all lie within known finite ranges. Condition (ii) is enforced by standard regularization in classifier training. Conditions (iii) and (iv) hold for all classifiers and cost functions used in our experiments.

### F.3. Proposition: Local Stability of Stackelberg Equilibrium

To analyze stability, we introduce the local best-response mappings of the two players. Given an equilibrium $(\theta^*, b^*)$, suppose there exist neighborhoods $\mathcal{U} \subset \Theta$ of $\theta^*$ and $\mathcal{V}$ of $b^*$ (in an appropriate function space) such that:

- The follower's best response is locally single-valued and continuously differentiable in $\theta$, defining a mapping $\Phi : \mathcal{U} \to \mathcal{V}$, $\theta \mapsto b_\theta$, where $b_\theta(x) = \Phi(\theta)(x)$;

- The leader's best response to a given follower mapping $b$ is locally single-valued and continuously differentiable in $b$, defining a mapping $\Psi : \mathcal{V} \to \mathcal{U}$, $b \mapsto \theta_b$.

Since $(\theta^*, b^*)$ is a Stackelberg equilibrium, we have $b^* = \Phi(\theta^*)$ and $\theta^* = \Psi(b^*)$. Define the composed mapping:

$$T := \Psi \circ \Phi : \mathcal{U} \to \mathcal{U}, \qquad T(\theta) = \Psi(\Phi(\theta)). \tag{65}$$

Note that $\theta^*$ is a fixed point of $T$:

$$T(\theta^*) = \Psi(\Phi(\theta^*)) = \Psi(b^*) = \theta^*. \tag{66}$$

**Proposition F.3** (Local Stability). *Let $(\theta^*, b^*)$ be a Stackelberg equilibrium under Pro-SF, and let $T = \Psi \circ \Phi$ be the composed mapping defined in Eq. (65). Suppose $T$ is locally contractive on $\mathcal{U}$: there exists $q \in (0, 1)$ such that for all $\theta_1, \theta_2 \in \mathcal{U}$,*

$$\|T(\theta_1) - T(\theta_2)\| \leq q \|\theta_1 - \theta_2\|. \tag{67}$$

*Then $(\theta^*, b^*)$ is locally stable: for any initialization $\theta_0 \in \mathcal{U}$, the best-response iteration*

$$\theta_{t+1} = T(\theta_t), \qquad b_t := \Phi(\theta_t), \tag{68}$$

*satisfies $\theta_t \to \theta^*$ and $b_t \to b^*$ as $t \to \infty$, with geometric rate:*

$$\|\theta_t - \theta^*\| \leq q^t \|\theta_0 - \theta^*\|. \tag{69}$$

*Proof.* We establish the result in four steps.

**Step 1: $\theta^*$ is a fixed point of $T$.** Since $(\theta^*, b^*)$ is a Stackelberg equilibrium:

$$b^* = \Phi(\theta^*), \qquad \theta^* = \Psi(b^*). \tag{70}$$

Therefore:

$$T(\theta^*) = (\Psi \circ \Phi)(\theta^*) = \Psi(b^*) = \theta^*. \tag{71}$$

Hence $\theta^*$ is a fixed point of $T$ in $\mathcal{U}$.

**Step 2: Convergence of the leader by Banach fixed-point theorem.** Since $T : \mathcal{U} \to \mathcal{U}$ is a contraction with constant $q \in (0, 1)$ on the complete metric space $\mathcal{U}$ (as a closed subset of $\mathbb{R}^d$), the Banach fixed-point theorem guarantees:

(a) $T$ has a *unique* fixed point in $\mathcal{U}$, which is $\theta^*$;

(b) For any $\theta_0 \in \mathcal{U}$, the iteration $\theta_{t+1} = T(\theta_t)$ converges geometrically:

$$\|\theta_t - \theta^*\| \leq q^t \|\theta_0 - \theta^*\| \to 0 \quad \text{as } t \to \infty. \tag{72}$$

**Step 3: Convergence of the follower.** Since $\Phi : \mathcal{U} \to \mathcal{V}$ is continuous (by the assumption that the follower's best response is continuously differentiable in $\theta$), the geometric convergence $\theta_t \to \theta^*$ implies:

$$b_t := \Phi(\theta_t) \to \Phi(\theta^*) = b^* \quad \text{as } t \to \infty. \tag{73}$$

**Step 4: Conclusion.** Combining Steps 2 and 3, we have $(\theta_t, b_t) \to (\theta^*, b^*)$ geometrically as $t \to \infty$, for any initialization $\theta_0 \in \mathcal{U}$. This establishes that the best-response dynamics return to $(\theta^*, b^*)$ from any sufficiently close initialization, i.e., the Stackelberg equilibrium is locally stable. $\qquad\square$

**Remark 7.** The contraction condition (67) is interpretable in terms of the underlying problem structure: it holds when the manipulation cost is sufficiently strong relative to the gain from manipulation in a neighborhood of the equilibrium, so that small perturbations to the classifier induce only small perturbations to agents' responses, and vice versa. This is a standard sufficient condition for stability in Stackelberg games and leader-follower systems (Li & Sethi, 2017).

### F.4. Corollary: Local Uniqueness of Stackelberg Equilibrium

**Corollary F.4** (Local Uniqueness). *Under the conditions of Proposition F.3, $(\theta^*, b^*)$ is the unique Stackelberg equilibrium in $\mathcal{U}$.*

*Proof.* Suppose $(\theta, b)$ is any Stackelberg equilibrium contained in $\mathcal{U} \times \mathcal{V}$. By follower optimality, $b = \Phi(\theta)$. By leader optimality, $\theta = \Psi(b)$. Substituting,

$$\theta = \Psi(b) = \Psi(\Phi(\theta)) = T(\theta). \tag{74}$$

Hence $\theta$ is a fixed point of $T$ in $\mathcal{U}$. By Step 2(a) of Proposition F.3, the Banach fixed-point theorem guarantees that $T$ has a unique fixed point in $\mathcal{U}$, which is $\theta^*$. Therefore $\theta = \theta^*$, and consequently:

$$b = \Phi(\theta) = \Phi(\theta^*) = b^*. \tag{75}$$

Hence $(\theta, b) = (\theta^*, b^*)$, establishing that $(\theta^*, b^*)$ is the unique Stackelberg equilibrium in $\mathcal{U} \times \mathcal{V}$. $\qquad\square$

## G. Additional Experiment

### G.1. Dataset

We evaluate our framework on five datasets, including four real-world and one synthetic benchmarks:

- **Credit** (Yeh, 2009): Credit card default prediction based on financial records.

- **Adult** (Becker & Kohavi, 1996): Income classification from census features.

- **Diabetes** (Teboul, 2015): A medical dataset containing clinical and demographic attributes for diabetes risk assessment.

- **German** (Hofmann, 1994): Credit risk classification based on personal and financial profiles.

- **Spam** (Hopkins et al., 1999): Email spam detection based on textual and statistical features.

- **Synthetic** (Lopez-Rojas et al., 2016): Simulated mobile transaction data for fraud detection.

### G.2. Notation Table

In the modeling process of Section 5, we designed a series of formulas and parameters. Here, we provide a notation table (Tab. 4) to facilitate a clearer understanding of our Pro-SF.

We preprocess each dataset following standard practice (categorical encoding, normalization, and train/validation/test splits).

*Table 3.* Summary of the public datasets used in our experiments.

| Dataset | #Features | #Instances | #Classes |
|---|---|---|---|
| Adult (Becker & Kohavi, 1996) | 14 | 48,842 | 2 |
| Credit (Yeh, 2009) | 23 | 30,000 | 2 |
| German (Statlog) (Hofmann, 1994) | 20 | 1,000 | 2 |
| Spambase (Hopkins et al., 1999) | 57 | 4,601 | 2 |
| CDC Diabetes (Teboul, 2015) | 21 | 253,680 | 2 |
| Synthetic (PaySim) (Lopez-Rojas et al., 2016) | 10 | 6,362,620 | 2 |

*Table 4.* Notation of prospect-guided utility and default experimental values.

| Symbol | Meaning | Default Value (Experiment) |
|---|---|---|
| $\alpha, \beta$ | Curvature parameters (diminishing sensitivity) | $\alpha = 0.8, \ \beta = 0.7$ |
| $\kappa$ | Loss aversion coefficient | 2.25 |
| $\gamma$ | Probability distortion parameter | 0.7 |
| $r$ | Reference point (discretized probability) | $\{0, 0.2, 0.4, 0.6, 0.8\}$ |
| $w(p)$ | Probability weighting functions | Inverse-S function (Eq. (10)) |
| $c(\mathbf{x}', \mathbf{x})$ | Manipulation cost function | Mahalanobis distance |

## G.3. Implementation Details

All experiments are conducted on a single NVIDIA TITAN $V$ (24GB) GPU. We implement the classifier $f$ as a logistic regression model trained with cross-entropy loss and learning rate $10^{-3}$. In prospect-theoretic utility, we set the curvature parameters $\alpha = 0.8$ and $\beta = 0.7$, the loss aversion coefficient $\kappa = 2.25$, and the probability-weighting parameter $\gamma = 0.7$ by default. In the mixed agent behavior paradigm, the proportion is set as $\pi = 0.2$. This choice reflects a realistic scenario where only a minority of agents behave in a fully rational manner, while the majority exhibit behavioral biases (Tversky & Kahneman, 1992; Ariely & Jones, 2008). Ablation studies are conducted on the *Credit* and *Synthetic* datasets under the *mixed* agent regime, representing real-world and controlled settings. To ensure robustness, we perform 10-fold cross-validation on all datasets.

In simulating agents' real-world manipulation process, we argue that agents need to exhibit heterogeneous behavioral biases. To capture this diversity, we randomly divide agents into several subgroups (e.g., three or four groups), and assign each subgroup different parameter combinations. This design mimics the fact that individuals in reality adopt distinct subjective strategies when manipulating their features. By contrast, when training the classifier to anticipate manipulations, we adopt a fixed parameter setting across all agents. This stabilizes the optimization process and ensures fair comparison across experiments, while still allowing evaluation against heterogeneous agent behaviors at deployment.

> For example, in one experimental run, the agents are randomly split into three groups. The first group is assigned ($\kappa = 2.00, \ \gamma = 0.70$), the second group ($\kappa = 2.25, \ \gamma = 0.75$), and the third group ($\kappa = 1.80, \ \gamma = 0.65$).

## G.4. More Ablation Studies

**More Ablation results** of our parameter analysis (including $\pi, r, (\alpha, \beta), \kappa,$ and $\gamma$) are shown in Table 5 and Fig. 6.

**An alternative of probability distortion.** The two-parameter *Prelec* function (Prelec, 1998) is :

$$w(p) \ = \ \exp\big(-\eta(-\ln p)^\phi\big), \qquad \phi, \eta > 0. \tag{76}$$

Compared to the one-parameter form used in the main text, the Prelec function provides greater flexibility. The parameter $\phi$ controls the shape of the curve: when $\phi < 1$, the function takes an inverse-$S$ form (small probabilities are given too much weight and large probabilities too little), while $\phi > 1$ produces an $S$-shape (the opposite pattern). In line with behavioral evidence, we restrict to the case $\phi < 1$, which produces the inverse-$S$ form. The parameter $\eta$ adjusts how strong this distortion is overall, with larger $\eta$ leading to a stronger down-weighting of probabilities across the board.

The ablation experimental results are summarized in Table 6.

*Table 5.* Overall accuracy (%) of rational-based classifiers and prospect-guided models across datasets under different $\pi$ in mixed behavior.

| Classifier | Mixed $\pi$ | Datasets | | | | | |
|---|---|---|---|---|---|---|---|
| | | *Adult* | *Credit* | *Diabetes* | *German* | *Spam* | *Synthetic* |
| *Rational-based* | *0.1* | $73.96_{\pm1.55}$ | $71.84_{\pm1.90}$ | $66.51_{\pm1.62}$ | $70.10_{\pm1.88}$ | $78.85_{\pm1.71}$ | $72.88_{\pm1.70}$ |
| | *0.2* | $74.31_{\pm1.51}$ | $72.12_{\pm1.87}$ | $66.81_{\pm1.65}$ | $70.43_{\pm1.92}$ | $79.16_{\pm1.63}$ | $73.15_{\pm1.74}$ |
| | *0.4* | $74.53_{\pm1.57}$ | $72.37_{\pm1.85}$ | $67.05_{\pm1.68}$ | $70.64_{\pm1.95}$ | $79.42_{\pm1.59}$ | $73.41_{\pm1.73}$ |
| *Pro-SF (ours)* | *0.1* | $\mathbf{78.92}_{\pm1.74}$ | $\mathbf{79.32}_{\pm1.66}$ | $\mathbf{72.23}_{\pm1.55}$ | $\mathbf{76.45}_{\pm1.68}$ | $\mathbf{86.02}_{\pm1.61}$ | $\mathbf{82.60}_{\pm1.58}$ |
| | *0.2* | $\mathbf{78.68}_{\pm1.77}$ | $\mathbf{79.13}_{\pm1.63}$ | $\mathbf{72.01}_{\pm1.58}$ | $\mathbf{76.24}_{\pm1.71}$ | $\mathbf{85.71}_{\pm1.52}$ | $\mathbf{82.34}_{\pm1.56}$ |
| | *0.4* | $\mathbf{78.50}_{\pm1.79}$ | $\mathbf{78.92}_{\pm1.65}$ | $\mathbf{71.84}_{\pm1.60}$ | $\mathbf{76.08}_{\pm1.73}$ | $\mathbf{85.38}_{\pm1.56}$ | $\mathbf{82.14}_{\pm1.57}$ |

*Table 6.* Accuracy comparison between the normalized power function and the Prelec probability weighting across multiple datasets under different parameter settings.

| Weighting | Datasets | | | | | |
|---|---|---|---|---|---|---|
| | *Credit* | *Adult* | *Diabetes* | *German* | *Spam* | *Synthetic* |
| $\gamma = 0.65$ | $78.68_{\pm1.25}$ | $79.13_{\pm1.10}$ | $72.01_{\pm1.05}$ | $76.24_{\pm1.20}$ | $85.71_{\pm1.18}$ | $82.34_{\pm1.15}$ |
| Prelec ($\phi = 0.65$, $\eta = 1.00$) | $78.50_{\pm1.28}$ | $79.00_{\pm1.12}$ | $71.80_{\pm1.08}$ | $76.10_{\pm1.18}$ | $85.42_{\pm1.21}$ | $82.10_{\pm1.14}$ |
| $\gamma = 0.70$ | $78.78_{\pm1.22}$ | $79.22_{\pm1.09}$ | $72.12_{\pm1.06}$ | $76.31_{\pm1.17}$ | $85.89_{\pm1.16}$ | $82.27_{\pm1.13}$ |
| Prelec ($\phi = 0.68$, $\eta = 1.00$) | $78.70_{\pm1.24}$ | $79.10_{\pm1.10}$ | $71.96_{\pm1.07}$ | $76.20_{\pm1.15}$ | $85.60_{\pm1.19}$ | $82.15_{\pm1.12}$ |
| $\gamma = 0.80$ | $78.82_{\pm1.20}$ | $79.24_{\pm1.08}$ | $72.18_{\pm1.05}$ | $76.33_{\pm1.15}$ | $86.02_{\pm1.15}$ | $82.12_{\pm1.11}$ |
| Prelec ($\phi = 0.75$, $\eta = 1.00$) | $78.79_{\pm1.22}$ | $79.20_{\pm1.09}$ | $72.05_{\pm1.06}$ | $76.30_{\pm1.14}$ | $85.78_{\pm1.17}$ | $82.08_{\pm1.10}$ |

### G.5. Results Analysis

Additional parameter analysis results are summarized in Table 5 and Fig. 6, covering the mixture ratio $\pi$, the reference point $r$, the weighting parameters $(\alpha, \beta)$, the loss-aversion factor $\kappa$, and the probability distortion parameter $\gamma$. As shown in Table 5, the proposed Pro-SF consistently outperforms the rational baseline across datasets, and the performance advantage holds for different proportions of mixed behavioral agents.

As shown in Fig. 6, when the discrete reference points are set to four or five bins, the accuracy remains stable, showing that the framework is robust under reasonably fine discretizations. However, when only three bins are used, we observe a slight performance drop. This is because with only three bins, the reference points become too sparse, forcing many agents with different true self-assessments to be grouped into the same anchor. As a result, the model cannot capture the finer granularity of subjective evaluations, and the induced manipulations are less accurately represented. This mismatch slightly weakens the behavioral modeling and leads to a small drop in accuracy.

Moreover, Fig. 6 show ablations under two different $(\alpha, \beta)$ settings. In both cases, the accuracy curves remain stable, indicating that the results are not sensitive to moderate shifts of these parameters. Across a wide range of $\kappa$ and $\gamma$ values, accuracy fluctuations are small (within $\pm0.3\%$), confirming that the framework is not overly dependent on fine-tuning these behavioral parameters.

Through extensive experimental design, we identified parameter settings of the normalized power function and the Prelec function that yield comparable weighting curves. As shown in Table 5, when evaluated under these matched configurations, the resulting accuracies across datasets are nearly identical. For example, results with $r = 0.7$ are largely similar to results with $\phi = 0.68, \eta = 1.0$. This means our framework can work with both types of probability distortion, as long as the parameters are set to produce comparable shapes.

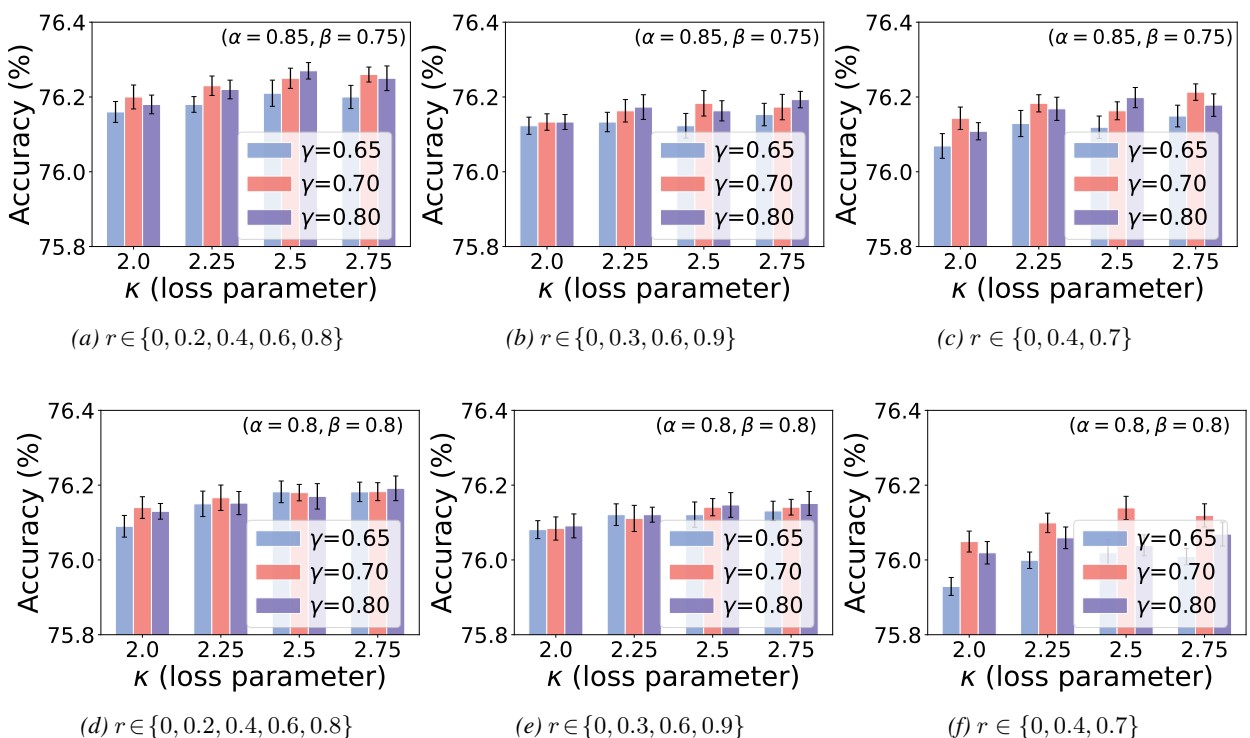

*Figure 6.* Parameter ablation results with parameters $\kappa, \gamma, r$ with different $(\alpha, \beta)$.

## H. Validation with Real World Manipulation Data

In this section, we provide further empirical support for our behavioral modeling. Recent work (Ebrahimi et al., 2025) conducted controlled human-subject experiments across several strategic-classification scenarios (e.g., hiring, medical decision-making). Their statistical findings show that:

- Human manipulation behavior **systematically deviates from the rational best-response assumption**;

- Individuals do not track the optimal boundary or follow theoretically optimal manipulation trajectories;

- Over-reaction and under-reaction behaviors appear frequently in practice.

These results provide direct evidence that **behavioral biases must be incorporated** into strategic-classification models, supporting the foundation of our Prospect-Based Utility.

To further validate our framework, we evaluate Pro-SF on two real human manipulation datasets released in (Ebrahimi et al., 2025): the job hiring dataset and the medical treatment dataset.

We compare two models:

- **Rational-SF**: the classical rational strategic-classification model;

- **Pro-SF**: our proposed Prospect-based strategic framework, behavioral parameters used in Pro-SF $\phi = \{\alpha = 0.78, \ \beta = 0.72, \ \kappa = 2.20, \ \gamma = 0.74\}$..

For each dataset, we report:

- **Classification Accuracy**;

- **Manipulation Deviation**: the average $\ell_2$ distance between real human manipulation $x \to x'$ and model-predicted manipulation $\hat{x}'$.

*Table 7.* Results on Human Manipulation Datasets: Job Hiring and Medical Treatment.

| Job Hiring Dataset | | | Medical Treatment Dataset | | |
|---|---|---|---|---|---|
| Model | Accuracy ↑ | Manip. Dev. ↓ | Model | Accuracy ↑ | Manip. Dev. ↓ |
| Rational-SF | 68.4% | 0.297 | Rational-SF | 71.9% | 0.267 |
| **Pro-SF** | **79.8%** | **0.148** | **Pro-SF** | **78.3%** | **0.181** |

Across both datasets, Pro-SF consistently achieves higher predictive accuracy, lower Manipulation deviation, and more stable alignment with human behavior.

These findings indicate that **Pro-SF closely matches actual human manipulation patterns**, offering strong empirical support for its validity.

## I. Learnability of Behavioral Parameters

This appendix studies whether the behavioral parameters $\phi = (\alpha, \beta, \kappa, \gamma)$ in the prospect-based utility can be *identified from observed strategic manipulation behavior*. We show that these parameters can be reliably inferred from both real human manipulation data and heterogeneous agent populations, and that classifier performance remains stable under estimation noise.

### I.1. Inverse Behavioral Inference via Discrete Choice

Given observed manipulation pairs $(x_i, x_i')$, we estimate the behavioral parameters by solving an inverse decision problem. Rather than assuming agents optimize over the full continuous feature space, we adopt a bounded rationality perspective and assume that agents choose among a finite set of salient manipulation options.

Formally, for each original feature vector $x$, we construct a candidate manipulation set $\mathcal{C}(x)$ consisting of feasible local manipulations (including $x$ itself). The likelihood of observing a manipulation $x_i'$ is modeled using a softmax (Boltzmann) choice rule:

$$P_\phi(x_i' \mid x_i) = \frac{\exp\big(\tau\, U_\mathrm{P}(x_i, x_i'; \phi)\big)}{\sum\limits_{x'' \in \mathcal{C}(x_i)} \exp\big(\tau\, U_\mathrm{P}(x_i, x''; \phi)\big)}, \tag{77}$$

where $\tau > 0$ controls decision stochasticity.

This formulation follows standard practice in inverse decision modeling, random utility models, and quantal response equilibria, and reflects that agents consider only a limited set of plausible manipulation options rather than optimizing globally.

The behavioral parameters are then inferred via maximum likelihood estimation:

$$\phi^* = \arg\max_\phi \sum_i \log P_\phi(x_i' \mid x_i). \tag{78}$$

### I.2. Learning Parameters from Real Human Manipulation Data

We first evaluate whether the behavioral parameters can be learned from real human decision data. We use two real-world manipulation datasets introduced in (Ebrahimi et al., 2025): a **job hiring** task and a **medical treatment** task.

For each dataset, we estimate $\hat{\phi}$ using the likelihood objective above and evaluate the resulting Pro-SF classifier. Table 8 reports the learned parameters and classification accuracy.

These results demonstrate that prospect-based behavioral parameters are *identifiable from real human manipulation trajectories*, supporting that Pro-SF does not rely on hand-tuned assumptions.

*Table 8.* Learned behavioral parameters and accuracy on real human manipulation datasets.

| Dataset | Learned Parameters $\hat{\phi}$ | Accuracy (%) |
|---|---|---|
| Job Hiring | $\{\alpha{=}0.79,\ \beta{=}0.73,\ \kappa{=}2.18,\ \gamma{=}0.72\}$ | **79.8** |
| Medical Treatment | $\{\alpha{=}0.81,\ \beta{=}0.70,\ \kappa{=}2.25,\ \gamma{=}0.71\}$ | **78.3** |

### I.3. Heterogeneous Strategic Populations

To further test robustness, we consider mixed agent populations containing both rational and behavioral agents. For a given proportion $\pi$ of rational agents:

- **Behavioral agents** $(1-\pi)$: each agent draws individual parameters $\phi_i$ from a distribution $\mathcal{P}_\phi$ and selects manipulations according to a noisy prospect-based decision:

$$x'_i = \arg\max_{x'} U_{\mathrm{P}}(x_i, x'; \phi_i) + \epsilon_i, \quad \epsilon_i \sim \mathcal{N}(0, \sigma^2). \tag{79}$$

- **Rational agents** $(\pi)$: agents follow the classical strategic classification model, maximizing acceptance probability minus manipulation cost.

We evaluate settings $\pi \in \{0.2, 0.4, 0.6\}$, corresponding to increasing dominance of rational behavior.

### I.4. Learned Parameters and Performance

Tables 9, 10, and 11 report the learned parameters $\hat{\phi}$ and the corresponding post-gaming accuracy achieved by the classifier trained with the estimated behavioral model.

*Table 9.* Learned parameters and accuracy with $\pi = 0.2$ (20% rational agents).

| Dataset | Learned Parameters $\hat{\phi}$ | Accuracy (%) |
|---|---|---|
| Adult | $\{\alpha = 0.78,\ \beta = 0.72,\ \kappa = 2.20,\ \gamma = 0.74\}$ | 81.42 |
| Credit | $\{\alpha = 0.80,\ \beta = 0.70,\ \kappa = 2.15,\ \gamma = 0.71\}$ | 82.30 |
| German | $\{\alpha = 0.81,\ \beta = 0.70,\ \kappa = 2.25,\ \gamma = 0.72\}$ | 79.21 |
| Synthetic | $\{\alpha = 0.79,\ \beta = 0.71,\ \kappa = 2.28,\ \gamma = 0.73\}$ | 85.05 |

*Table 10.* Learned parameters and accuracy with $\pi = 0.4$ (40% rational agents).

| Dataset | Learned Parameters $\hat{\phi}$ | Accuracy (%) |
|---|---|---|
| Adult | $\{\alpha = 0.78,\ \beta = 0.74,\ \kappa = 2.20,\ \gamma = 0.74\}$ | 81.32 |
| Credit | $\{\alpha = 0.80,\ \beta = 0.70,\ \kappa = 2.10,\ \gamma = 0.70\}$ | 82.20 |
| German | $\{\alpha = 0.80,\ \beta = 0.72,\ \kappa = 2.18,\ \gamma = 0.71\}$ | 79.25 |
| Synthetic | $\{\alpha = 0.77,\ \beta = 0.75,\ \kappa = 2.20,\ \gamma = 0.72\}$ | 84.91 |

Tables 9–11 report the inferred parameters and post-manipulation accuracy. Across all values of $\pi$, the estimated parameters remain stable and yield consistent classification performance.

These results indicate that: (i) behavioral parameters are learnable from heterogeneous strategic behavior; (ii) Pro-SF is robust to parameter estimation noise; and (iii) accurate modeling does not require access to agents' true latent parameters.

## J. Behavioral Illustration

We provide two illustrative contexts that highlight these mechanisms and motivate our formulation.

*Table 11.* Learned parameters and accuracy with $\pi = 0.6$ (60% rational agents).

| Dataset | Learned Parameters $\hat{\phi}$ | Accuracy (%) |
|---|---|---|
| Adult | $\{\alpha = 0.78,\ \beta = 0.76,\ \kappa = 2.05,\ \gamma = 0.70\}$ | 81.28 |
| Credit | $\{\alpha = 0.78,\ \beta = 0.74,\ \kappa = 2.03,\ \gamma = 0.68\}$ | 82.16 |
| German | $\{\alpha = 0.80,\ \beta = 0.73,\ \kappa = 2.10,\ \gamma = 0.69\}$ | 79.08 |
| Synthetic | $\{\alpha = 0.78,\ \beta = 0.75,\ \kappa = 2.15,\ \gamma = 0.70\}$ | 84.84 |

## J.1. Financial Investment.

Consider two individuals facing the same investment opportunity: investing $10,000 with a 60% chance of gaining $20,000 and a 40% chance of losing the entire principal.

A classical rational utility model predicts that both investors should take the action because the expected payoff is positive. In practice, however, behavior diverges. An individual with $100,000 in savings may choose to invest, whereas another with only $20,000 may refrain, even though the expected value is identical.

This discrepancy arises because the two individuals have different *reference points*: a $10,000 loss represents 10% of the former's wealth but 50% of the latter's. When combined with loss aversion, the same monetary loss induces a substantially larger psychological cost for the low-wealth individual. The Prospect-Based Utility accounts for this effect through the reference-point term $r$ and the asymmetric loss-weighting parameter $\kappa$, leading to distinct optimal decisions for the two agents.

## J.2. Disease Screening.

In health-risk environments, individuals often display behavior that cannot be explained by classical rational models. Even when the true prevalence of a disease is extremely low (e.g., 0.1%), people may repeatedly undergo medical testing or stockpile medication.

Such behavior reflects probability distortion: very small probabilities are overweighted and treated as non-negligible risks. Moreover, external events (e.g., alarms, exposure, or stress) may shift one's psychological reference point from "I am healthy" to "I might already be at risk," sharply increasing the perceived cost of inaction. Under classical SC models, a 0.1% risk should induce only minimal adjustment.

In contrast, Prospect-Based Utility naturally captures this form of overreaction through the weighting function $w(p)$ and the dynamic reference point $r$.

Together, these examples illustrate behavioral patterns that classical rational utilities cannot encode, motivating the need for a prospect-theoretic formulation in strategic classification.

