# OpenReview forum: "Beyond Rational Illusion: Behaviorally Realistic Strategic Classification"
_ICML.cc/2026/Conference — ICML 2026 regular_

### Official Review · Reviewer_fj22 · 2026-03-10

**Soundness:** 2
**Presentation:** 3
**Significance:** 2
**Originality:** 3
**Overall Recommendation:** 4
**Confidence:** 3

**Summary:**

This paper studies strategic classification when agents are not perfectly rational. It introduces the problem of behaviorally realistic strategic classification and proposes Pro-SF, a prospect theory inspired framework that models agent responses using three ingredients: loss aversion, reference bias, and probability distortion. The paper also argues that training under a purely rational response model can fail through two qualitatively different modes. Empirically, the paper gives results on six datasets

**Compliance With Llm Reviewing Policy:**

Affirmed.

**Final Justification:**

The rebuttal addressed the key concerns raised. They added some missing assumptions which accurately reflect the (reduced) strength of their results. I have increased my score slightly.

**Key Questions For Authors:**

- Eqn 12: can you explain why this form is better than simpler stochastic or bounded rationality response models?
- How is Eq. (13)’s manipulation argmax solved in practice?
- Can the main theoretical propositions be restated with the actual assumptions used in the proofs, rather than stronger unconditional wording?

**Limitations:**

yes

**Strengths And Weaknesses:**

Strengths:
- The paper targets a real limitation of standard strategic-classification models
- It is a meaningful direction for the area, and the motivating examples in the introduction are intuitive and compelling.
- The paper is generally readable, figures are particularly helpful.

Weaknesses:
- Theoretical claims appear stronger than the proofs actually establish: for instance, Prop 4.1 in the main text reads as if any mismatch $D_\theta \neq D_\theta^\mathrm{rat}$ implies positive deployment error, but the appendix proof (eqn 20) adds an extra assumption. Props 4.3 and 4.5 also presented broadly in main text, but appendix proofs require additional conditions e.g., $\tau_A > 2\varepsilon_R$ and $\tau_B > 2\varepsilon_R$.
- Regarding eqn 12, the authors do not convincingly derive why this exact form is the right translation of prospect theory into strategic classification. I would have liked a stronger argument for why this parameterization should be preferred over simpler models.
- The experimental setting seems favorable to the proposed model. The non-rational and mixed agents appear to be generated using essentially the same behavioral ingredients that Pro-SF assumes. That makes the gains in Table 1 plausible, but it also raises the concern that the evaluation partly rewards model-match.
- The main comparison is largely against a rational SC baseline. For a paper whose central claim is that rational agent modeling is insufficient, I think it is important to compare against other non-rational or bounded-rationality models

Comments/Typos:
- Line 320: “is inferred” -> are inferred
- Line 357: “five datasets” but after that lists six
- Line 357: “other forms included in Appendix K” -> doesn’t seem correct?

---

> ### Author Rebuttal · Authors · 2026-03-31
>
> Dear Reviewer fj22,
>
> Thank you for your thoughtful comments and suggestions. We would like to address your concerns point by point.
>
> > [W2&Q1]: Justify why Eqn 12 is preferred over simpler stochastic/bounded rationality models.
>
> **Response:**
>
> We thank the reviewer for this question.
> - Eq. (12) is not intended as the only translation of Prospect Theory, but as a **behaviorally grounded instantiation** that preserves its key mechanisms in strategic classification.
> - Compared to simpler stochastic or bounded-rationality models, which mainly treat deviations as noise or imperfect optimization, Eq. (12) explicitly captures **systematic and asymmetric behavioral patterns**, including reference dependence, loss aversion, and probability weighting.
> - In strategic settings, agents’ responses are not random, but are shaped by gain/loss framing around decision thresholds and distorted perceptions of success. Such asymmetric behavior is not well captured by noise-based models.
>
> Therefore, our formulation is not just more flexible, but **better aligned with empirically observed decision behavior**, providing a more interpretable and realistic response model.
>
> > [W1&Q3] Restated propositions with the actual assumptions used in the proofs
>
> **Response:**
> Thanks for the suggestion! Due to the limited space, we have restated **propositions 4.1, 4.3 &4.5, and made detailed discussions of these assumptions in the following comments**.
>
> Also, we have revised our paper with the restated propositions.
>
> > [W3]: Experimental setting favors Pro-SF due to generating non-rational/mixed agents using the same behavioral assumptions.
>
> **Response:**
>
> We thank the reviewer for this important observation regarding potential model-match bias.
> - We would like to clarify that our original experimental setup is not designed to favor Pro-SF.
>   - The non-rational agents are generated based on behavioral principles from Prospect Theory, which are widely used to model human decision-making in social settings.
>   - In addition, we consider **mixed populations** consisting of both rational and non-rational agents to better reflect realistic settings.
> - To further address this concern, **we expand the current experimental evaluation with two additional experiments**: one using alternative response models and one using real temporal datasets.
>     - Due to space constraints, **we place the full results and discussion in the following comments**.
>
> > [W4]: Compare against other non-rational or bounded-rationality models beyond the rational baseline.
>
> **Response:**
> - To address this concern, we incorporate four representative non-rational response formulations inspired by prior work, and compare them directly with Pro-SF under the same temporal-data evaluation setting. Specifically, we adopt the response modeling paradigms from:
>   - [1]: noisy utility response;
>   - [2]: nonlinear utility-based response;
>   - [3]: environment-aware response;
>   - [4]: random-cost response.
> - We report results on two temporal datasets: *OULAD* and *NLSY97*.
>
> **Table 4. Results on OULAD.**
> |Model|Accuracy(%)|
> |-|-|
> |Rational baseline|77.84|
> |[1]|81.34|
> |[2]|81.08|
> |[3]|81.53|
> |[4]|80.97|
> |**Pro-SF(ours)**|**83.26**|
> |
>
> **Table 5. Results on NLSY97.**
> |Model|Accuracy(%)|
> |-|-|
> |Rational baseline|73.12|
> |[1]|77.98|
> |[2]|78.72|
> |[3]|80.21|
> |[4]|79.65|
> |**Pro-SF(ours)**|**81.68**|
> |
> - **Analysis.** Across both datasets, Pro-SF outperforms not only the rational SC baseline, but also several representative bounded-rationality models. This suggests that the advantage of Pro-SF comes from its ability to capture structured behavioral deviations more effectively.
>
> > [Q2]: How is Eq. (13)’s manipulation argmax solved in practice?
>
> **Response:**
>
> In practice, the argmax in Eq. (13) is implemented through a **discrete candidate-based approximation**.
> - For each input $x$, we first construct a finite candidate set $C(x)$ over the actionable features by discretizing the feasible manipulation space according to feature bounds and cost constraints.
> - For each candidate $x' \in C(x)$, we evaluate the prospect-based utility $U_P(x,x')$ using the current classifier output $p(x')$, together with the corresponding value transformation, probability weighting, and manipulation cost.
> - We then approximate the argmax using a softmax (Boltzmann) rule:
>   $$
>   P(x' \mid x)=\frac{\exp(\tau U_P(x,x'))}{\sum_{z\in C(x)}\exp(\tau U_P(x,z))},
>   $$
>   where $\tau$ is a temperature parameter. As $\tau$ increases, this distribution concentrates on higher-utility candidates and approaches the argmax solution.
> - The resulting computation is linear in the candidate set size, i.e., $O(|C(x)|)$.
>
> **We have added these experiments and discussions to our revised paper.**
>
> ---
> [1] Alternative microfoundations for strategic classification
>
> [2] Non-linear welfare-aware strategic learning
>
> [3] Causal strategic learning with competitive selection
>
> [4] Learning the distribution map in reverse causal performative prediction

---

> > ### Author Rebuttal · Reviewer_fj22 · 2026-04-01
> >
> > I thank the authors for the clarification, and I hope they will make the revisions.

---

> > > ### Author Response · Authors · 2026-04-01
> > >
> > > Dear Reviewer fj22,
> > >
> > > We appreciate your feedback. Due to space constraints, some details in our responses to **[W1&Q3] and [W3]** were condensed.
> > >
> > > **We provide additional clarifications and experiments below to help verify that our revisions fully address your concerns**.
> > >
> > > > [W1&Q3] Restated propositions with the actual assumptions used in the proofs.
> > >
> > > **Response:** Thank you for these important suggestions. In the revision, we have restated these propositions so that the main-text statements match the assumptions used in the proofs.
> > >
> > > Specifically, we have revised:
> > > - **Lines 193-201: We restate Proposition 4.1:**
> > >
> > > Let $f_\theta^{rat}$ denote a classifier optimized on $D_\theta^{\text{rat}}$, and define
> > > $$
> > > \delta(\theta):= \mathbb{E}[\mathcal{L}(f_\theta^{rat}(x), y) ].
> > > $$
> > > **If $D_\theta \neq D_\theta^{\text{rat}}$, and that this mismatch affects a non-zero mass of samples near the decision boundary, then** $\delta(\theta) > 0$, i.e., a non-vanishing deployment error persists.
> > >
> > > - **Lines 214-219: We restate Proposition 4.3:**
> > >
> > > Let $f_R$ be a classifier trained under the rational assumption. **If the error induced by over-defense is sufficiently large compared to the residual rational error** (e.g., $\tau_A > 2\epsilon_R$), **then** :
> > > $$
> > > \mathbb{E}[\mathcal{L}(f_R(x), y)]>\mathbb{E}[\mathcal{L}(f_R(b_R(x)), y)],
> > > $$
> > > where $\mathcal{L}$ is the loss function.
> > >
> > > - **Lines 227-234: We restate Proposition 4.5:**
> > >
> > > Let $f_R$ be a classifier trained under the rational assumption. **If the error induced by under-defense is sufficiently large compared to the residual rational error (e.g., $\tau_B > 2\epsilon_R$), then**
> > > $$
> > > \mathbb{E}[\mathcal{L}(f_R(x^*), y)]>\mathbb{E}[\mathcal{L}(f_R(b_R(x)), y)],
> > > $$
> > > where $\mathcal{L}$ is the loss function.
> > >
> > > > [W3]: Experimental setting favors Pro-SF due to generating non-rational agents using the same behavioral assumptions.
> > >
> > > **Response:** To fully address this concern, we have conducted two additional experiments that explicitly remove this alignment.
> > >
> > > **1.  Evaluation under external response models (model misspecification).**
> > >
> > > - We evaluate Pro-SF under a set of **external response models** adopted from prior work, including:  Noisy Response[1], Nonlinear Response[2], and Competitive Response [3].
> > > - These response mechanisms are structurally different from the prospect-theoretic formulation, and therefore constitute clear **model misspecification settings**.
> > > - We compare Pro-SF against the standard rational baseline under these externally generated manipulations.
> > >
> > > **Table 1. Accuracy (%) under external response models.**
> > > |Response Model|Predictive Model|Accuracy(%)|
> > > |-|-|-|
> > > |Noisy|Rational Baseline|81.24|
> > > ||Pro-SF(ours)|**83.76** |
> > > |Nonlinear|Rational Baseline |79.08|
> > > ||Pro-SF(ours)|**83.69**|
> > > |Competitive Response|Rational Baseline|79.77|
> > > ||Pro-SF(ours)| **82.83**|
> > > |
> > >
> > > - **Analysis.** Pro-SF consistently outperforms the rational baseline across all external response settings, indicating that the gains are not driven by model alignment but reflect robustness to diverse and realistic behaviors.
> > >
> > > **2. Evaluation on real data.**
> > > - To further reduce reliance on simulated responses, we consider a setting based on **real temporal data**, i.e., OULAD[4], NLSY97[5].
> > > - Specifically, we treat $(x_t, y)$ as pre-response samples and $(x_{t+1}, y)$ as post-response samples, forming a natural **before-after response setting**.
> > > - Models are evaluated on $(x_{t+1}, y)$.
> > >
> > > **Table 2. Accuracy (%) under real temporal transitions.**
> > > |Dataset|Predictive Model|Accuracy(%)|
> > > |-|-|-|
> > > |OULAD |Rational Baseline|77.84|
> > > ||Pro-SF(ours)|**83.26**|
> > > |NLSY97|Rational Baseline|74.12|
> > > ||Pro-SF(ours)|**81.78**|
> > > |
> > >
> > > - **Analysis.** Pro-SF consistently outperforms the rational baseline on longitudinal data, suggesting that its advantage does not rely on matching a specific simulated response model.
> > >
> > > We have added these experiments to our revised paper.
> > >
> > > > [Some Typos]
> > >
> > > **Response:** We have revised all typos, including
> > > - are inferred (Line 320);
> > > - six datasets (Line 357);
> > > - multi-agent forms in Appendix K (Line 357).
> > >
> > > **We thank the reviewer for the helpful comments, which have improved the quality of our work. All revisions have been incorporated into the revised paper.**
> > >
> > > ----
> > > [1] Jagadeesan, Meena, Celestine Mendler-Dünner, and Moritz Hardt. Alternative microfoundations for strategic classification. ICML, 2021.
> > >
> > > [2] Xie, Tian, and Xueru Zhang. Non-linear welfare-aware strategic learning. Proceedings of the AAAI/ACM Conference on AI, Ethics, and Society.  2024.
> > >
> > > [3] Vo, Kiet QH, et al. Causal strategic learning with competitive selection. Proceedings of the AAAI Conference on Artificial Intelligence. 2024.
> > >
> > > [4] Kuzilek, Jakub, Martin Hlosta, and Zdenek Zdrahal. "Open University Learning Analytics dataset. UCI Machine Learning Repository, 2015
> > >
> > > [5] Michael, Robert T., and Michael R. Pergamit. The National Longitudinal Survey of Youth, 1997 Cohort. The Journal of Human Resources, vol. 36, no. 4, 2001, pp. 628–40. JSTOR

---

### Official Review · Reviewer_VhhW · 2026-03-11

**Soundness:** 2
**Presentation:** 2
**Significance:** 2
**Originality:** 2
**Overall Recommendation:** 4
**Confidence:** 3

**Summary:**

The authors define the behaviorally realistic strategic classification problem, where agents' strategic manipulations deviate from full rationality due to psychological biases. They propose the Prospect-Guided Strategic Framework (Pro-SF) to address the problem of strategic manipulations in three special cases: loss aversion, reference bias, and probability distortion.

**Compliance With Llm Reviewing Policy:**

Affirmed.

**Final Justification:**

The authors fully answered my questions and provided evidence about the changes. I raised my score to a positive rate.

**Key Questions For Authors:**

1. Changes in notation are often confusing. For example, $b_R(x)$ is used in Equation (1) and $T_{\theta}(x)$ in Equation (3) (similarly for the utility, which appears in Equation (1) and as $U$ in Equation (3)). In Equation (13), the notation switches back to $\hat{b}$.

2. In Equation (3), $\theta$ is not defined, and how does $\theta$ affect $U$?

3. $D_{\theta}$ is not mathematically defined. Is it a function $(x,y) \mapsto D_{\theta}(x,y)$? Or is $D_{\theta} = T_{\theta}$, i.e., simply a map transforming $x$?

4. In Example 1, what does $\theta$ represent? Is it $w$? Similarly, what is $\theta$ in Proposition 4.3 and in Example 2? In definition 5.2. is $\theta = (\alpha,\beta,\kappa,\gamma)$? If so, are they identifiable (in eq 15 you write $\phi$ to be equal of the maximum argument of the likelihood, so I believe you assume they are identifiable)?

5. Proposition 4.1 appears to rely on the assumption that the discrepancy between the ex-ante and ex-post distributions occurs in a decision-relevant region near the boundary (lines 704–705). However, this assumption does not seem to be explicitly stated in the proposition.

6. Similarly to Question 5, Proposition 4.3 appears to rely on the assumption that $P(A) > 0$, but this assumption does not seem to be explicitly stated in the proposition.

7. Some papers address the problems of non-fully rational agents by using random utility. For example, in [1], the agents' cost is random, allowing them to be not fully rational, and the distribution of the cost is estimated. How does this relate to the present work?

### Typos

1. Line 774 should be \mathbf{1}.

### References

[1] Bracale, D., Maity, S., Banerjee, M., & Sun, Y. (2024). *Learning the distribution map in reverse causal performative prediction.* arXiv preprint arXiv:2405.15172.

**Limitations:**

The limitations of the methodology are not stated. E.g. the paper provides an algorithm for the special cases of loss aversion, reference bias, and probability distortion: is it possible to design an algorithm for a more general behavioural manipulation?

**Strengths And Weaknesses:**

**Soundness and presentation**: The idea of modeling agents’ strategic manipulation as a deviation from full rationality due to biases (such as psychological ones) is an interesting direction. However, the authors often mix or change notation, which makes the paper hard to follow (see Questions 1, 2, 3, and 4 in the Questions section). In addition, some theorems do not appear to be fully stated (see Questions 5 and 6 in the Questions section). The experiments appear to be well conducted.

**Significance and originality**: Relevant references related to non–full rational agents are missing (see Question 7 in the Questions section), where the agent’s utility is modeled as random rather than deterministic, which in turn makes the agent’s decision random. Moreover, the paper provides an algorithm for the special cases of loss aversion, reference bias, and probability distortion: Is it possible to design an algorithm for a more general behavioural manipulation?

Overall, the significance of the work appears to be reasonable, but in my opinion, the paper would benefit from clearer and more precise definitions and statements.

---

> ### Author Rebuttal · Authors · 2026-03-31
>
> > Question 1-4: Notation clarification.
>
> **Response:** We thank the reviewer for these careful comments. We agree that the notation in the current version is not sufficiently consistent, and **we have revised it to make each part fully explicit (a full revised version is included in the following comments due to the limited space)**.
> - We unify all response-related mappings under a single notation family $b(\cdot)$:
>   - $b_R(x)$: rational strategic manipulation;
>   - $b_\theta(x)$: general behavioral strategic manipulation;
>   - $b_P(x)$: prospect-guided strategic manipulation in Pro-SF.
> - Accordingly, we revise Eq. (3) and Eq. (13) as
>   $$
>   b_\theta(x)\in \arg\max_{x'\in\mathcal X} U_\theta(x,x'),
>   \quad
>   b_P(x)\in \arg\max_{x'\in\mathcal X} U_P(x,x';\phi).
>   $$
> - We also make the role of parameters explicit:
>   - $\theta$ denotes a **general behavioral parameterization** used in Section 4;
>   - $\phi$ denotes the **Prospect-Theoretic parameter set** in Pro-SF, e.g. $(\alpha,\beta,\kappa,\gamma)$, used in Section 5.
> - Finally, we clarify that the post-manipulation object is a **distribution**, not a function, by defining
>   $$
>   (X,Y)\sim\mathcal D,\quad X'=b_\theta(X),\quad \mathcal D_\theta:=\mathcal L(X',Y),
>   $$
>   where $\mathcal D_\theta$ is the distribution induced by the behavioral response $b_\theta$.
>
> These revisions also remove notation switches such as $T_\theta$, $\hat b$, and $b_B$, and make the relationship among the response rule, utility, parameters, and induced distribution consistent throughout the paper.
>
> > **Question 5:** However, this assumption does not seem to be explicitly stated in the proposition.
> **Response** We have restated the proposition with explicit assumptions, and provide a detailed discussion of these assumptions in the following comments.
>
> **Proposition 4.1 (Deployment bias under behavioral mismatch).**
> Let $ f_R $ denote a classifier optimized on $ \mathcal{D}_R $, and define the deployment error:
>
> $$
> \delta(\theta) := \mathbb{E}_{(x,y)\sim \mathcal{D}_\theta}\big[\mathcal{L}(f_R(x),y)\big] \ge 0,
> $$
> where $ \mathcal{L} $ is the loss function.
>
> If the mismatch between $\mathcal{D}_\theta$ and $\mathcal{D}_R$ occurs with nonzero mass in a decision-relevant region near the classifier boundary, and this mismatch changes the induced prediction risk on that region, then optimizing under the rational-agent assumption leads to a non-vanishing deployment error, i.e.,
> $$
> \delta(\theta) > 0.
> $$
>
> > **Question 6:** Similarly to Question 5, Proposition 4.3 appears to rely on the assumption
>
> **Response** We have restated the proposition with explicit assumptions, and provide a detailed discussion of these assumptions in the following comments.
>
> **Proposition 4.3 (Accuracy Degradation from Over-defense).**
> Under over-defense, suppose there exists a nonzero-probability set of samples for which agents do not carry out the rationally anticipated manipulation $b_R(x)$, and as a result the classifier’s defensive shift changes their decisions from acceptance under the anticipated response to rejection at deployment. Then a classifier trained under the rational-agent assumption achieves lower accuracy, i.e.,
> $$
> \delta(\theta):=\mathbf{E}[\mathcal{L}(f_R(x),y)] \ge 0,
> $$
> where $\mathcal{L}$ is the loss function for classification.
>
> > **Question 7:** Some papers address the problems of non-fully rational agents by using random utility.
>
> **Response:**
>
> We thank the reviewer for pointing out this insightful work. We agree that modeling non-fully rational agents via random utility is a closely related direction.
>
> - **Relation to our work.**
>   - Both frameworks relax the fully rational-agent assumption and aim to better capture realistic strategic behavior beyond deterministic best response.
> - **Key difference.**
>   - The random-utility framework[1] models deviations from rationality mainly through **stochastic variability**, e.g., random perturbations to costs.
>   - Our framework instead models **systematic and behaviorally grounded deviations**, motivated by Prospect Theory. In particular, these deviations are not symmetric random noise: they are structured distortions such as loss aversion, reference dependence, and asymmetric probability weighting.
>
> - **Why this matters.**
>   - Random utility is well-suited to capturing unobserved heterogeneity or noisy decision-making.
>   - Our framework is designed to capture **realistic behavioral biases** that are especially natural in social decision settings, where responses are often shaped by psychologically grounded asymmetries rather than purely symmetric randomness.
>
> - **Summary.**
>   - We therefore view the two perspectives as complementary: random utility emphasizes stochastic variation, while our work emphasizes structured behavioral bias.
>   - Incorporating both stochastic and prospect-theoretic components would be a promising direction for future work.
>
>
> **We have revised the paper and added these new propositions.**

---

> > ### Author Rebuttal · Reviewer_VhhW · 2026-04-03
> >
> > Thank you for your clarifications. I raise my score assuming the authors will make the revisions they promised.

---

> > > ### Author Response · Authors · 2026-04-03
> > >
> > > Dear Reviewer VhhW,
> > >
> > > **We appreciate your feedback and helpful suggestions, which have improved the quality of our work.**
> > >
> > > We would like to **provide more details for your questions to help you verify that our revisions fully address your concerns**.
> > >
> > > > **Question 1-4:** Notation clarification and revision.
> > >
> > > **Supplement**
> > > - In the revised manuscript, we have made the notation fully consistent and clarified the roles of all parameters.
> > > - To make our revision clear, we would like to provide a detailed list of the corresponding manuscript revisions below:
> > >
> > > **Table 1. Revision Table**
> > > |Location|Original Content|Revised Content|Note|
> > > |-|-|-|-|
> > > |Lines 177–181 (left)|$T_\theta : x \mapsto x' \in \arg\max_{x'} U_\theta(x,x')$|$b_\theta(x) \in \arg\max_{x' \in \mathcal X} U_\theta(x,x')$|Eq.(3), unify notation for behavioral response|
> > > |Lines 183–190 (left)|$D_\theta$, $D_\theta^{\mathrm{rat}}$|$(X,Y)\sim\mathcal D, X' = b_\theta(X), \mathcal D_\theta:=\mathcal L(X',Y)$| Explicitly define induced post-manipulation distribution|
> > > |Lines 195–200 (left)|$D_\theta$, $D_\theta^{\mathrm{rat}}$|$\mathcal D_\theta$, $\mathcal D_R$||
> > > |Lines 236–242 (left)|$b_B(x)$|$b_\theta(x)$|Replace inconsistent notation|
> > > |Lines 289–299 (left)|$U_P(x,x')$|$U_P(x,x';\phi)$| Make dependence on Prospect-Theoretic parameters explicit |
> > > |Lines 311–318 (right)|$\hat b_P(x_i)$, $\hat b_P(x)$|$b_P(x_i)$, $b_P(x)$||
> > > |Lines 320–324 (right)|$\hat b_P(x) \in \arg\max_{x'} U_P(x,x')$|$b_P(x) \in \arg\max_{x' \in \mathcal X} U_P(x,x';\phi)$| Consistent Pro-SF response definition|
> > > |Lines 324–328 (right)|$\mathcal{L}(f(\hat b_P(x)), y)$|$\mathcal{L}(f(b_P(x)), y)$ | Consistent learning objective|
> > > |Lines 326–329 (right)|$\phi=\\{\alpha,\beta,\kappa,\gamma\\}$|$\theta$: general behavioral parameter ; $\phi$: Prospect-Theoretic parameters in Pro-SF|Clarify parameter roles|
> > > |Line 774|tpyos|$\mathbf{1}$||
> > > |
> > >
> > > > **Question 5&6:** Restated propositions with the actual assumptions.
> > >
> > > **Supplement**
> > >
> > > In the revision, we have restated the propositions so that the main-text statements match the assumptions used in the proofs and added discussions on corresponding assumptions in the Appendix.
> > >
> > > Specifically, we have revised:
> > > - **Lines 193-201: We restate Proposition 4.1:**
> > >
> > > Let $f_\theta^{rat}$ denote a classifier optimized on $D_\theta^{rat}$, and define the deployment error
> > >
> > > $$
> > > \delta(\theta):=\mathbb{E}\_{(x,y)\sim D_\theta}[\mathcal{L}(f_\theta^{rat}(x),y)].
> > > $$
> > >
> > > **If $D_\theta \neq D_\theta^{rat}$, and this mismatch places non-zero mass in a decision-relevant region near the decision boundary, then** $\delta(\theta)>0$, i.e., a non-vanishing deployment error persists.
> > >
> > > - **In Appendix B, lines 731-733: we add a discussion** on the assumption of Proposition 4.1.
> > >     - In classification settings, only changes near the decision boundary can alter predicted labels and thus affect outcomes. Deviations far from the boundary typically leave the decision unchanged and have negligible impact on deployment error.
> > >     - Therefore, requiring the mismatch to place non-zero mass in a decision-relevant boundary region is a mild and realistic assumption. In practice, whenever behavioral deviations affect outcomes, this condition naturally holds.
> > >
> > > - **Lines 214-219: We restate Proposition 4.3:**
> > >
> > > Let $f_R$ be a classifier trained under the rational assumption. **If the over-defense region $A$ has non-zero probability mass**, and for these samples the classifier’s defensive shift turns them from accepted under the anticipated response to rejected at deployment, then
> > > $$
> > > \mathbb{E}[\mathcal{L}(f_R(x), y)]>\mathbb{E}[\mathcal{L}(f_R(b_R(x)), y)],
> > > $$
> > > where $\mathcal{L}$ is the loss function.
> > >
> > > - **In Appendix C, lines811-815: we add a discussion** on the assumption of Proposition 4.3.
> > >     - In practice, agents may not follow the manipulation predicted by the rational model due to uncertainty or behavioral frictions, especially near the decision boundary where gains are marginal. As a result, over-defensive classifiers may incorrectly reject such agents.
> > >     - Therefore, assuming that these deviations occur on a set with non-zero probability mass is mild and realistic. In practical settings, behavioral frictions commonly prevent a fraction of agents from acting as predicted, making this condition naturally satisfied.
> > >
> > > > **Question 7:** Discussion on Related work.
> > >
> > > **Supplement**
> > > - **In Section 2, lines 113–115, we add a discussion** of the related work [1].
> > >   - Random-utility models[1] introduce stochastic perturbations to agents’ costs, allowing deviations from fully rational behavior through random utility.
> > >   - Our work models systematic and behaviorally grounded deviations, focusing on structured biases such as loss aversion, reference dependence, and probability distortion.
> > >
> > > ----
> > >
> > > **Thank you again for the helpful comments.**
> > >
> > > **In the revised version of our paper, we have carefully incorporated all your suggestions to further improve the quality of our work.**

---

### Official Review · Reviewer_FgHH · 2026-03-13

**Soundness:** 3
**Presentation:** 3
**Significance:** 3
**Originality:** 3
**Overall Recommendation:** 5
**Confidence:** 3

**Summary:**

The paper claims that existing strategic classification frameworks typically assume rational agents. The authors then propose a new framework that allows for more complex agents' behaviour. Specifically, the new framework incorporates the three key mechanisms from prospect theory to model different aspects of agents' complex behaviour.

They show that classifiers that anticipate agents' behaviour based on this rich response model can outperform classifiers that rely on the standard agents' rational behavoural model.

**Compliance With Llm Reviewing Policy:**

Affirmed.

**Final Justification:**

The rebuttal addressed my concerns. In particular, the authors provided
- a clear justification of how the three irrational aspects are combined, with a real-world example;
- a clarification on what specific information the agents possess that makes them act according to this prospect-based model;
- an elaboration on the contrast between their agents' behavioural model and existing formulations;
- more extensive experiment results to investigate the learnability of agents' model's parameters.

I raise my score to 5, and strongly encourage the authors to incorporate their clarifications and additional results to the revision.

**Key Questions For Authors:**

1. Can the authors provide a justification for the combination of components in Definition 5.2? And give a real-world example on how those components interact with each other?

2. Can the authors discuss more the learnability of behavioural parameters? In particular, how many distinctive classifiers should one deploy to obtain some guaranteed result?

3. Can the authors discuss existing relaxations of the agents' response model in prior work more? And if your framework can incorporate those models?

**Limitations:**

The paper does not discuss its limitations. I think the paper could benefit from an elaboration on how future work can tackle the learnability of behavioural parameters.

**Strengths And Weaknesses:**

Soundness: The paper is sound overall, but I have two main concerns:
- The three irrational mechanisms are well discussed and explained on page 5, but not their combination in Definition 5.2. The authors did not provide any justification for why those three components are combined this way. For example, they could give a real-world example on how these components interact and discuss more on why other forms of combinations are not considered.
- The discussion of the learnability of behavioural parameters is a bit too brief. It is unclear to me how many distinctive classifiers one has to deploy to collect sufficiently rich data for inferring these parameters. This is because data about agents' responses is specific to the deployed classifier and without enough variation, it is unclear how the learner can infer the agents' model.

Presentation: The paper is generally well written and clear.

Significance: The paper tackles an important problem by making the agents' response model richer. The paper approaches this from a principled way by utilising ideas from prospect theory and their proposed response model looks general enough.

Originality: The originality of the paper is a bit concerning, due to the lack of discussion of many relevant works in strategic machine learning that already generalise the agents' response model to some extent. The paper could benefit from discussing the models of those works in more detail to highlight their distinctions. For example, some works that consider non-linear and heterogeneous cost functions, noisy agents' responses (Jagadeesan et al. 2021, Harris et al., 2022, Vo et al., 2024, Xie and Zhang, 2024).


References:
- Jagadeesan, Meena, Celestine Mendler-Dünner, and Moritz Hardt. "Alternative microfoundations for strategic classification." International Conference on Machine Learning. PMLR, 2021.
- Harris, Keegan, et al. "Strategic instrumental variable regression: Recovering causal relationships from strategic responses." International Conference on Machine Learning. PMLR, 2022.
- Vo, Kiet QH, et al. "Causal strategic learning with competitive selection." Proceedings of the AAAI Conference on Artificial Intelligence. Vol. 38. No. 14. 2024.
- Xie, Tian, and Xueru Zhang. "Non-linear welfare-aware strategic learning." Proceedings of the AAAI/ACM Conference on AI, Ethics, and Society. Vol. 7. No. 1. 2024.

---

> ### Author Rebuttal · Authors · 2026-03-31
>
> Dear Reviewer FgHH,
>
> Thank you for your thoughtful comments and suggestions. We would like to address your concerns point by point.
>
> > [Concern1/Q1]: Justify the combination of the three irrational mechanisms in Definition 5.2.
>
> **Response:**
>
> - The combination in Definition 5.2 follows the standard structure of Prospect Theory and is adapted to the strategic classification setting. The key idea is that the three components capture **distinct and complementary roles in decision-making**, rather than being arbitrarily combined:
>   - **Reference dependence** defines how the agent evaluates the current state relative to a threshold (i.e., gain vs. loss region).
>   - **Loss aversion** determines the asymmetric valuation of outcomes once this reference is fixed.
>   - **Probability weighting** distorts the perceived likelihood that a manipulation will lead to a favorable outcome.
>
> - Together, they form a coherent behavioral pipeline: agents first **frame** their position, then **evaluate outcomes asymmetrically**, and finally **distort success probabilities** when deciding whether and how to manipulate.
>
> - **A concrete example is job screening:**
>   - An applicant slightly below the interview cutoff perceives this threshold as a **reference point** (near-miss framing), experiences strong motivation to avoid rejection due to **loss aversion**, and may **overweight the small probability** that minor resume changes will push them above the cutoff. As a result, the applicant may invest disproportionate effort in strategic adjustments.
>
> > [Concern2/Q2]: Discuss learnability of behavioral parameters.
>
> **Response:**
> - We agree that collecting response data under a single deployed classifier may be insufficient to reliably estimate behavioral parameters, since the observed responses are classifier-specific.
>
> - **Clarification on learnability.**
>   - In our setting, behavioral parameters are inferred from agents’ manipulation pairs under the deployed decision rule.
>   - In practice, parameter learnability mainly depends on the diversity of deployed classifiers: **more diverse responses lead to more robust parameter estimation**.
>
> - **Supplementary experiment.**
>   - To examine this point empirically, we introduce **multiple classifiers with diverse structures and evaluate how classifier diversity affects parameter estimation**.
>   - Specifically, we generate agents’ responses under classifiers that differ in (i) decision threshold, (ii) model parameters (e.g., weight vectors in linear classifiers), and (iii) margin / acceptance region.
>   - We then vary the number of such distinct classifiers used for data collection, estimate the behavioral parameters using our method.
>   - We report both the learned parameters and accuracy.
>
>   **Table 1. Learned parameters and accuracy**
>   |# Classifiers|Learned Parameters $\hat{\phi}$|Accuracy (%)|
>   |-|-|-|
>   |1|{α=0.72, β=0.69, κ=2.05, γ=0.68}|81.84±0.35|
>   |3|{α=0.76, β=0.70, κ=2.14, γ=0.71}|82.33±0.27|
>   |5|{α=0.79, β=0.72, κ=2.21, γ=0.73}|82.84±0.18|
>   |8|{α=0.80, β=0.73, κ=2.24, γ=0.74}|83.21±0.14|
>   |10|{α=0.80, β=0.73, κ=2.25, γ=0.74}|83.28±0.13|
>
> - **Analysis.**
>   - The results show that increasing classifier diversity improves parameter recovery and downstream performance, with gains gradually saturating.
>   - Empirically, a small number of distinct classifiers already provides strong estimates, while additional classifiers mainly yield diminishing refinements.
>
> > [Concern3/Q3]: Discuss existing relaxations of response model.
>
>  **Response:**
>
> We thank the reviewer for this insightful comment.
> - **Relation to prior work.**
>   - Existing works extend strategic classification by enriching the response model, such as introducing noisy responses [1], causal formulations [2,3], or more general nonlinear cost structures [4].
>   - **Our perspective.** Rather than increasing the functional flexibility of the response model, we introduce a **behaviorally grounded and structured response model** based on Prospect Theory. This formulation captures **asymmetric and distorted decision patterns**, leading to more realistic modeling of agent behavior beyond standard rational or noise-based assumptions.
> - These prior response models can be viewed as specific forms of bounded or non-fully rational behavior. **We provide additional experimental results under such settings in the following comments.**
>
>
>
> We have added these experiments and discussions to our revised paper.
>
> ----
> References:
>
> [1] Jagadeesan, Meena, Celestine Mendler-Dünner, and Moritz Hardt. "Alternative microfoundations for strategic classification."
>
> [2] Harris, Keegan, et al. "Strategic instrumental variable regression: Recovering causal relationships from strategic responses."
>
> [3] Vo, Kiet QH, et al. "Causal strategic learning with competitive selection."
>
> [4] Xie, Tian, and Xueru Zhang. "Non-linear welfare-aware strategic learning."
>
> [5] Moritz Hardt,Nimrod Megiddo,Christos Papadimitriou,et al. Strategic Classification.

---

> > ### Author Rebuttal · Reviewer_FgHH · 2026-04-03
> >
> > Thank you for the clarification. I have some comments and a follow-up question related to Q3.
> >
> > [Concern1/Q1] I understand this now. I will consider adjusting my score. I suggest the authors incorporate this clarification into their revision and additionally provide a brief discussion of your utility model's weaknesses.
> >
> > [Concern2/Q2] This result seems very positive. Please consider discussing what scenarios would make it more challenging to learn the behavioural parameters with limited number of deployed classifiers. This would give interesting insights into this problem.
> >
> > [Concern3/Q3] My follow-up question is that: what kind of information about the classifier that the agents possess that makes them act this way (i.e., maximising the prospect utility-based model)?
> >
> > To elaborate, many works in strategic machine learning motivate their modelling choices by explicitly saying what the agents know about the decision maker's classifier. For instance, classical works like that of Hardt et al. (2015) consider the case where agents have full knowledge of the classifier; in Bechavod et al. (2022), agents know only historical data, hence they infer the classifier using ERM; in Cohen et al. (2024), agents know partial description of the classifier and perform Bayesian update; in Xie & Zhang (2024), agents only have access to local approximates of the actual classifier.
> >
> > Given that the agents in your work have probability distortion and reference bias, I suppose they do not have access to the classifier and cannot optimise it directly. Then, what information about the classifier do they actually possess? Is it a realistic scenario?
> >
> > References:
> > - Hardt, Moritz, et al. "Strategic classification." Proceedings of the 2016 ACM conference on innovations in theoretical computer science. 2016.
> > - Bechavod, Yahav, et al. "Information discrepancy in strategic learning." International Conference on Machine Learning. PMLR, 2022.
> > - Cohen, Lee, et al. "Bayesian strategic classification." Advances in Neural Information Processing Systems 37 (2024): 111649-111678.
> > - Xie, Tian, and Xueru Zhang. "Non-linear welfare-aware strategic learning." Proceedings of the AAAI/ACM Conference on AI, Ethics, and Society. Vol. 7. No. 1. 2024.

---

> > > ### Author Response · Authors · 2026-04-04
> > >
> > > Dear Reviewer VhhW,
> > >
> > > We appreciate your feedback and helpful suggestions, which have improved the quality of our work.
> > >
> > > To better organize our responses, we first address Q3, followed by Q2 and Q1.
> > >
> > > > **[Concern 3/Q3] Information available to agents.**
> > >
> > > **Response:** We thank the reviewer for this important question.
> > > - We would like to clarify that, in our setting, **agents do not have access to the current classifier, but they have full knowledge of past data.**
> > >   - Concretely, agents have access to historical feedback and outcomes, but do not know the exact form of the current classifier when making decisions.
> > >   - Therefore, they do not optimize the classifier directly. They form subjective beliefs about their likelihood of success based on past observations, and act according to this perceived success probability.
> > > - **This setting reflects realistic scenarios.**
> > >   - For example, in credit approval systems, applicants may observe that increasing income or reducing existing debt improved past approval outcomes, but they do not know the exact decision rule used by the current model.
> > >   - Similarly, in hiring, candidates may infer from past experience that certain skills or credentials improve their chances, without knowing how these features are weighted by the current screening system.
> > >   - **In both cases, individuals rely on past outcomes to form subjective beliefs, and their decisions are further shaped by psychological factors such as reference-dependent evaluation and distorted perceptions of success.**
> > > - **Relative to prior work:** our setting lies between full-information and no-information assumptions: agents do not directly observe the current classifier, but they are not completely uninformed either, since they can rely on historical outcomes to guide their decisions.
> > >
> > >
> > > > **[Concern 2/Q2]**
> > >
> > > We thank the reviewer for this insightful suggestion. We agree that in realistic settings, learning behavioral parameters becomes more challenging.
> > > - We consider the two following scenarios:
> > >     - **Scenario 1: limited variation across deployed classifiers.** When only a few classifiers are deployed and their decision rules are similar, agents face nearly identical incentives, resulting in limited variation in observed manipulation behavior.
> > >     - **Scenario 2: limited agent awareness of the classifier.** In some settings, agents have little or no knowledge of the decision rule (e.g., only result or no feedback), similar to work[1], so observed behavior mixes intrinsic preferences with uncertainty, making parameter learning more difficult.
> > > - **Additional simulation experiments:** To further examine the case, we add a simulation study:
> > >     - In the parameter-learning stage, manipulation pairs are collected when agents have little knowledge of the classifier.
> > >     - We then use the learned parameters to predict in a standard classifier-aware setting.
> > >
> > > **Table 1. Performance Comparison.**
> > > |Setting|Learned Parameters|Accuracy|
> > > |-|-|-|
> > > |Known|{α=0.79,β=0.73,κ=2.18,γ=0.72}|82.73|
> > > |Unknown|{α=0.82,β=0.70,κ=2.24,γ=0.75}| 78.91|
> > > |
> > >
> > > - **Analysis.** As shown in Table 1, the learned parameters deviate more and the accuracy drops, confirming that these settings make behavioral parameter learning more challenging.
> > > - **We consider some directions to address these challenges.**
> > >   - One approach is to increase the diversity of observed responses, e.g., by leveraging data from more varied deployment settings or naturally evolving decision environments.
> > >   - Another direction is to incorporate uncertainty-aware or partially observed behavioral models, leading to more robust parameter estimation under sparse feedback.
> > >
> > > > **[Concern1/Q1]**
> > > - **In Section 5.2, lines 317–321: we added a remark** to clarify the combination of the three behavioral components.
> > >   - The remark explicitly states that the formulation in Eq. (12) follows the behavioral evaluation structure of Prospect Theory: agents first determine their position relative to a reference point, which defines the gain-loss framing; conditioned on this framing, they evaluate outcomes asymmetrically via loss aversion; and finally, they distort success probabilities when assessing the desirability of manipulation.
> > > - **In lines 322–325: we include a brief real-world example** (job screening) to illustrate how these components interact in practice.
> > > - **In lines 425-427: before Conclusion, we added a limitation discussion:**
> > >     - Our parameter learning relies on informative manipulation pairs collected from agent–classifier interactions. When these observations are limited, the estimated parameters may be weaker. Improving parameter learning under such low-information settings remains an important direction for future work.
> > >
> > > Thank you again for the helpful comments.
> > >
> > > **In the revised version of our paper, we have carefully incorporated all your suggestions to further improve the quality of our work.**
> > >
> > > ---
> > > [1]  Ghalme et al. ''Strategic Classification in the Dark.'' ICML 2021.

---

### Official Review · Reviewer_4igi · 2026-03-13

**Soundness:** 2
**Presentation:** 3
**Significance:** 3
**Originality:** 2
**Overall Recommendation:** 4
**Confidence:** 3

**Summary:**

This paper studies how to design a better decision-agent (people subjected to decisions from a model) reaction model. Previously these agent's utility functions are often too rational, in the sense they are simply argmax_{x' \in \mathcal{X}} f(x') - c(x,x'). The authors argue that this assumption is often unrealistic, as human decision making is influenced by behavioral biases. To address this limitation, the paper proposes incorporating three behavioral mechanisms inspired by prospect theory into the agent response model: (1) asymmetry in how individuals perceive gains and losses (loss aversion), (2) the role of subjective reference points, and (3) the tendency to overweight small probability events.

The authors analysed how deviations from rational-agent assumption can lead to over-defense or under-defense in deployment. Experimets on synthetic and real-world datasets, including ablation studies on different behavioral (hyper)parameters are provided.

**Compliance With Llm Reviewing Policy:**

Affirmed.

**Final Justification:**

I think the rebuttal have addressed my concerns, I have raised my scores from weak reject to weak accept.

**Key Questions For Authors:**

1. The proposed framework introduces several behavioural parameters (e.g., loss aversion, probability distortion, and reference dependence) into the agent response model. In practice, how would these parameters be reliably estimated or validated from real-world data? The paper mentions maximum likelihood estimation using observed manipulation pairs, but in many strategic classification settings such manipulation data may not be directly observable. Could the authors elaborate on how the model would be calibrated or deployed in more realistic settings?

2. The theoretical analysis mainly highlights deployment mismatch when training under rational-agent assumptions but encountering behaviourally biased agents at deployment. However, the current characterisation of over-defence and under-defence seems relatively straightforward. Are there deeper structural properties of the resulting Stackelberg game under prospect-theoretic utilities (e.g., equilibrium existence, uniqueness, or stability) that could be analysed? Such results might strengthen the theoretical contribution.

**Limitations:**

Yes

**Strengths And Weaknesses:**

# Soundness
This work proposes a behaviorally realistic extension of strategic classification by incorporating various mechanisms (loss aversion, reference dependence, and probability distribution) into the agent's reaction model to predictive model's outcome. The modelling framework is clearly specified and integrates these componetns into the standard Stackelberg interaction between the decision maker and strategic agents. Empirical evaluations are provided. However, the theory is relatively limited.. for example the under and over defence are quite simplisitic and obvious results (if you train on a distribution A but see distribution B, of course you will suffer in performance). Whereas a deeper characterisation of the resulting game or equilibrium properties of strategic classification under such settings would be of more general interest to the community.

# Presentation
Overall, the paper is clearly written and well structured. The motivation for relaxing the rational-agent assumption is well explained, and the behavioral mechanisms are illustrated with intuitive examples that help convey the core ideas. The narrative generally flows logically from problem formulation to modeling and experiments. That said, some modeling assumptions could be explained more clearly, particularly the interpretation of behavioral parameters and how they might be estimated or validated in practical settings besides simply saying maximum likelihood.

# Signifiance
The study of more realistic behavioural model is an important problem for strategic classification indeed.

# Originality
The main originality of the work lies in introducing the notion of behaviorally realistic strategic classification and integrating prospect-theoretic behavioral mechanisms into the agent response model. This provides an interesting conceptual extension of existing strategic classification frameworks. However, there are limited new algorithmic or theoretical insights in this work.

---

> ### Author Rebuttal · Authors · 2026-03-31
>
> Dear Reviewer 4igi,
>
> Thank you for your thoughtful comments and suggestions. We would like to address your concerns point by point.
> > **[Q1]:** Could the authors elaborate on how the model would be calibrated or deployed in more realistic settings?
>
> **Response:**
>
> We thank the reviewer for this important question.
> - In practice, obtaining manipulation pairs would require observing the same individuals twice, before and after their feature adjustments. We agree that in real-world settings, explicit manipulation pairs may not often be directly observable.
> - To address this, we adopt a **temporal calibration strategy** that leverages time-series data to approximate manipulation behavior.
>   - Specifically, we treat two consecutive observations of the same individual, $(x_t, x_{t+1})$, as an approximate manipulation pair.
> - **Experiment 1: temporal calibration**
>   - We use these temporal pairs to estimate the behavioral parameters, and then evaluate the resulting model on future unseen states (e.g., $(x_{t+2}, y)$, $(x_{t+3}, y)$).
>   - **Datasets.** We instantiate this procedure on real temporal datasets:
>     - **OULAD**[1]: a student learning dataset where features evolve over time as students adapt their study behavior;
>     - **NLSY97**[2]: a longitudinal socioeconomic dataset tracking individuals’ attributes across years.
> - **Results.** We report the learned parameters and accuracy:
>
>   **Table 1. Learned parameters and accuracy evaluated on $x_{t+2}$ and $x_{t+3}$.**
>   |Dataset|Learned Parameters $\hat{\phi}$| Accuracy on $x_{t+2}$ (%)|Accuracy on $x_{t+3}$ (%)|
>   |-|-|-|-|
>   |OULAD|{α = 0.79, β = 0.73, κ = 2.18, γ = 0.72}|80.84|80.21|
>   |NLSY97|{α = 0.81, β = 0.71, κ = 2.22, γ = 0.74}|82.15|81.63|
>   |
> - **Experiment 2: cross-dataset validation**
>   - To further examine whether the learned parameters are overly task-specific, we additionally use the parameters learned from one temporal dataset and directly evaluate them on the other dataset.
>   - We report the parameters and accuracy:
>
>   **Table 2. Cross-dataset validation of learned parameters.**
>   |Original Dataset|Validation Dataset|Learned Parameters$\hat{\phi}$|Accuracy (%)|
>   |-|-|-|-|
>   |OULAD|NLSY97|{α = 0.79, β = 0.73, κ = 2.18, γ = 0.72}|81.77|
>   |NLSY97|OULAD|{α = 0.81, β = 0.71, κ = 2.22, γ = 0.74}|79.92|
>  |
> - **Analysis:**
>   - These results show that temporal data provides a practical way to calibrate behavioral parameters even without explicitly observed manipulation pairs.
>   - The cross-dataset validation further indicates that Pro-SF is not overly sensitive to parameter values: parameters learned from one dataset remain effective on another and still outperform a rational classifier, consistent with the sensitivity results in Fig. 4 and Fig. 5 of our paper.
>
> > **[Q2]:** Are there deeper structural properties of the resulting Stackelberg game under prospect-theoretic utilities (e.g., equilibrium existence, uniqueness, or stability) that could be analysed?
>
> **Response:**
>
> We thank the reviewer for the insightful question and suggestions. To address this concern, we **have added two propositions** to inform the existence of Stackelberg Equilibrium in Pro-SF, local stability of Stackelberg Equilibrium, **and a corollary** to inform the local uniqueness of Stackelberg equilibrium, with their **proofs in our following comments later** due to limited space:
>
> - **Proposition (Existence of Stackelberg Equilibrium)**
>
> Under the Pro-SF framework, if the feasible action set of agents and the parameter space of the classifier are both compact, then the induced strategic game admits a Stackelberg equilibrium.
>
>
> - **Proposition (Local Stability of Stackelberg Equilibrium)**
>
> Let $(\theta^{\star}, b^* )$ be a Stackelberg equilibrium under the Pro-SF framework.
> If the manipulation cost is sufficiently strong in a neighborhood of the equilibrium, and the composed best-response mapping of the leader–follower system is locally contractive around $\theta^*$, then the Stackelberg equilibrium is locally stable.
>
>
> - **Corollary (Local Uniqueness of Stackelberg Equilibrium)**
>
> Under the conditions of the proposition on the local stability of Stackelberg equilibrium, $(\theta^*, b^*)$ is the unique Stackelberg equilibrium in a neighborhood of the equilibrium point.
>
> **We have revised the paper and added these new propositions and proofs.**
>
> ----
>
> Reference:
>
> [1] Kuzilek, Jakub, Martin Hlosta, and Zdenek Zdrahal. "Open University Learning Analytics dataset." UCI Machine Learning Repository, 2015
> [2] Michael, Robert T., and Michael R. Pergamit. “The National Longitudinal Survey of Youth, 1997 Cohort.” The Journal of Human Resources, vol. 36, no. 4, 2001, pp. 628–40. JSTOR

---

> > ### Author Rebuttal · Reviewer_4igi · 2026-04-03
> >
> > Thank you for taking my question 2 seriously and proposed some propositions on this regard. I would love to take a look at the proof, maybe you can reply here? Thanks.

---

> > > ### Author Response · Authors · 2026-04-03
> > >
> > > Dear Reviewer 4igi,
> > >
> > > **We appreciate your feedback and helpful comments, which have improved the quality of our work. We would like to provide the proofs for Question 2 below.**
> > >
> > > ---
> > > **Proof 1: Existence of Stackelberg Equilibrium**
> > >
> > > We prove the result in three steps.
> > > - Step 1: Existence of the follower's best response.
> > >
> > > Fix any $\theta \in \Theta$ and any $x$. When $\mathcal X$ is compact, and $U_P(x,x';\theta)$ is continuous in $x'$, since it is composed of continuous terms including the classifier score (or probability), the prospect-based value terms, the probability-weighting functions, and the manipulation cost.
> > >
> > > Therefore, by the Weierstrass extreme value theorem,
> > > $$
> > > \max_{x' \in \mathcal{X}}U_P(x,x'; \theta)
> > > $$
> > > admits a maximizer. Hence,
> > > $$
> > > \arg\max_{x'\in\mathcal{X}}U_P(x,x';\theta)\neq\varnothing,
> > > $$
> > > so the follower's best response exists.
> > > - Step 2: Existence of the leader's optimum.
> > >
> > > Given the follower's best response, define the induced leader objective
> > > $$
> > > F(\theta)=\mathbb E[L(f_\theta(b*(x;\theta)),y)].
> > > $$
> > > When $\Theta$ is compact, and both $f_\theta$ and $L$ are continuous in their arguments. Hence, $F(\theta)$ is continuous on $\Theta$. Applying the Weierstrass theorem, there exists $\theta* \in \Theta$ such that
> > > $$
> > > F(\theta*)=\min_{\theta \in \Theta} F(\theta).
> > > $$
> > > - Step 3: Existence of the equilibrium.
> > >
> > > Step 1 shows that for every $\theta\in\Theta$, the follower has a best response $b*(x;\theta)$. Step 2 shows that there exists $\theta* \in \Theta$ minimizing the leader's objective under these responses. Therefore, the pair $(\theta*,b*(\cdot;\theta*))$ satisfies:
> > >
> > > 1. $b*(x;\theta*)$ is an optimal response to $\theta*$;
> > > 2. $\theta*$ is optimal for the leader given the follower's response.
> > >
> > > Hence, $(\theta*,b*(\cdot;\theta*))$ constitutes a Stackelberg equilibrium.
> > >
> > > ---
> > > **Proof 2: Local Stability of Stackelberg Equilibrium**
> > >
> > > We proceed in four steps:
> > > - Step 1: Local best-response mappings of the follower and the leader.
> > >
> > > Given model parameters $\theta$, the follower solves
> > > $$
> > > b(x;\theta)\in \arg\max_{x'\in\mathcal X}U_P(x,x';\theta),
> > > $$
> > > where $U_P(x,x';\theta)$ is the prospect-based utility.
> > >
> > > If in a neighborhood of $(\theta*,b*)$, the follower problem admits a locally unique optimizer that depends continuously on $\theta$, the follower defines a local best-response mapping
> > > $$
> > > \Phi:\theta\mapsto b_\theta,\quad b_\theta(x)=\Phi(\theta)(x).
> > > $$
> > > Given a response function $b$, the leader solves
> > > $$
> > > \min_{\theta\in\Theta}J(\theta;b),\quad J(\theta;b)=\mathbb E[L(f_\theta(b(x)),y)].
> > > $$
> > > Similarly, in a neighborhood of the equilibrium, the leader problem admits a locally unique optimizer that depends continuously on $b$. Thus, the leader defines a local best-response mapping
> > > $$
> > > \Psi:b\mapsto\theta_b.
> > > $$
> > > Since $(\theta*,b*)$ is a Stackelberg equilibrium,
> > > $$
> > > b^*=\Phi(\theta*),\quad  \theta^*=\Psi(b*).
> > > $$
> > > - Step 2: Fixed point of the composed mapping.
> > >
> > > Define
> > > $$
> > > T(\theta):=(\Psi\circ\Phi)(\theta).
> > > $$
> > > Then, by the equilibrium conditions,
> > > $$
> > > T(\theta*)=\Psi(\Phi(\theta*))=\Psi(b*)=\theta*.
> > > $$
> > > Hence, $\theta*$ is a fixed point of $T$.
> > > - Step 3: Convergence of the leader updates.
> > >
> > > When $T$ is locally contractive in some neighborhood $\mathcal U$ of $\theta*$, i.e., there exists $q\in(0,1)$ such that for all $\theta_1,\theta_2\in\mathcal U$,
> > > $$
> > > \|T(\theta_1)-T(\theta_2)\|\le q \|\theta_1-\theta_2\|.
> > > $$
> > > By the Banach fixed-point theorem, $T$ has a unique fixed point in $\mathcal U$, and for any initialization $\theta_0 \in \mathcal U$, the iteration
> > > $$
> > > \theta_{t+1}=T(\theta_t)
> > > $$
> > > satisfies
> > > $$
> > > \theta_t\to\theta* \quad  \text{as } t\to\infty.
> > > $$
> > > - Step 4: Convergence of the follower responses.
> > >
> > > Since $\Phi$ is continuous near $\theta*$, the convergence $\theta_t\to\theta*$ implies
> > > $$
> > > b_t:=\Phi(\theta_t) \to \Phi(\theta*)=b*.
> > > $$
> > > Therefore,
> > > $$
> > > (\theta_t,b_t)\to(\theta*,b*).
> > > $$
> > >
> > > Hence, the best-response dynamics return to $(\theta*,b*)$ from any sufficiently close initialization, establishing the local stability of the Stackelberg equilibrium.
> > >
> > > ---
> > > **Proof 3: Local Uniqueness of Stackelberg Equilibrium**
> > >
> > > From Proof 2, the leader-follower system induces the composed mapping
> > > $$
> > > T(\theta):=(\Psi \circ \Phi)(\theta),
> > > $$
> > > and $\theta*$ satisfies $T(\theta*)=\theta*$, i.e., it is a fixed point of $T$.
> > >
> > > Moreover, $T$ is locally contractive in a neighborhood $\mathcal U$ of $\theta*$. Hence, by the Banach fixed-point theorem, $T$ admits a unique fixed point in $\mathcal U$, which is $\theta*$.
> > >
> > > Now, any Stackelberg equilibrium $(\theta,b)$ in $\mathcal U$ satisfy
> > > $$
> > > b=\Phi(\theta), \quad  \theta=\Psi(b),
> > > $$
> > > and thus
> > > $$
> > > \theta=(\Psi \circ \Phi)(\theta) = T(\theta).
> > > $$
> > > That is, $\theta$ must be a fixed point of $T$. By uniqueness, $\theta=\theta*$, and therefore
> > > $$
> > > b=\Phi(\theta*)=b*.
> > > $$
> > > Hence, $(\theta*,b*)$ is the unique Stackelberg equilibrium in a neighborhood of the equilibrium point.
> > >
> > > ---
> > > **Thanks again for your insightful suggestions. All revisions have been incorporated into the revised paper.**

---

### Decision · Program_Chairs · 2026-04-30

**Decision:**

Accept (regular)

**Comment:**

Strategic classification is usually studied under the assumption of fully rational agents.  This paper examines the implications of various behavioral biases.  Overall the reviewers and I ultimately appreciate the direction.  While the reviewers identified a number of weaknesses in the original submission, additional experiments and analysis introduced in the discussion were able to allay these concerns.  In the "final justifications" the reviewers acknowledged satisfaction with the proposed changes and updated scores accordingly. So while this paper will certainly not be the final word on this issue, I believe it contributes meaningfully to our understanding of what changes about strategic classification if agents are not fully rational.